# Gene expression profiling of patient-derived pancreatic cancer xenografts predicts sensitivity to the BET bromodomain inhibitor JQ1: implications for individualized medicine efforts

Benjamin Bian[1] (iD), Martin Bigonnet[1], Odile Gayet[1], Celine Loncle[1], Aurélie Maignan[1], Marine Gilabert[1], Vincent Moutardier[1,2,3], Stephane Garcia[1,2], Olivier Turrini[1,4], Jean-Robert Delpero[4], Marc Giovannini[4], Philippe Grandval[5], Mohamed Gasmi[2,3], Mehdi Ouaissi[5], Veronique Secq[2], Flora Poizat[4], Rémy Nicolle[6], Yuna Blum[6], Laetitia Marisa[6], Marion Rubis[1], Jean-Luc Raoul[4], James E Bradner[7], Jun Qi[7], Gwen Lomberk[8], Raul Urrutia[8], Andres Saul[9], Nelson Dusetti[1,*] (iD) & Juan Iovanna[1,**] (iD)

## Abstract

*c-MYC* controls more than 15% of genes responsible for proliferation, differentiation, and cellular metabolism in pancreatic as well as other cancers making this transcription factor a prime target for treating patients. The transcriptome of 55 patient-derived xenografts show that 30% of them share an exacerbated expression profile of MYC transcriptional targets (MYC-high). This cohort is characterized by a high level of Ki67 staining, a lower differentiation state, and a shorter survival time compared to the MYC-low subgroup. To define classifier expression signature, we selected a group of 10 MYC target transcripts which expression is increased in the MYC-high group and six transcripts increased in the MYC-low group. We validated the ability of these markers panel to identify MYC-high patient-derived xenografts from both: discovery and validation cohorts as well as primary cell cultures from the same patients. We then showed that cells from MYC-high patients are more sensitive to JQ1 treatment compared to MYC-low cells, in monolayer, 3D cultured spheroids and *in vivo* xenografted tumors, due to cell cycle arrest followed by apoptosis. Therefore, these results provide new markers and potentially novel therapeutic modalities for distinct subgroups of pancreatic tumors and may find application to the future management of these patients within the setting of individualized medicine clinics.

**Keywords** bromodomains; *c-MYC*; JQ1; pancreatic adenocarcinoma; transcriptomic signature
**Subject Categories** Cancer; Chromatin, Epigenetics, Genomics & Functional Genomics

## Introduction

Pancreatic ductal adenocarcinoma (PDAC) is one of the most lethal cancers and a major public health issue since there are approximately 230,000 new PDAC cases per year worldwide with approximately the same number of deaths (Jemal *et al*, 2005). Like others malignant diseases, PDAC results from a complex combination of genetic, epigenetic, and environmental factors which gives rise to a particularly heterogeneous disease, with patients having different set of symptoms, predisposition to early metastasis, and therapeutic responses (Yachida & Iacobuzio-Donahue, 2013; Dunne & Hezel, 2015; Waddell *et al*, 2015). This heterogeneity highlights the necessity to stratify patients with the goal of predicting better responses to therapies (Heller *et al*, 2015; Koay *et al*, 2016; Noll *et al*, 2016). One strategy to discover potential markers for patient stratification

1 Centre de Recherche en Cancérologie de Marseille (CRCM), INSERM U1068, CNRS UMR 7258, Parc Scientifique et Technologique de Luminy, Aix-Marseille Université and Institut Paoli-Calmettes, Marseille, France
2 Hôpital Nord, Marseille, France
3 CIC1409, AP-HM-Hôpital Nord, Aix-Marseille Université, Marseille, France
4 Institut Paoli-Calmettes, Marseille, France
5 Hôpital de la Timone, Marseille, France
6 Programme Cartes d'Identité des Tumeurs (CIT), Ligue Nationale Contre Le Cancer, Paris, France
7 Department of Medical Oncology, Dana-Farber Cancer Institute, Harvard Medical School, Boston, MA, USA
8 Laboratory of Epigenetics and Chromatin Dynamics, Departments of Biochemistry and Molecular Biology and Medicine, Mayo Clinic, Rochester, MN, USA
9 Centre Interdisciplinaire de Nanoscience de Marseille-CNRS UMR 7325, Parc Scientifique et Technologique de Luminy, Marseille, France
 *Corresponding author. Tel: +33 491 828828; Fax: +33 491 82886083; E-mail: nelson.dusetti@inserm.fr
 **Corresponding author. Tel: +33 491 828803; Fax: +33 491 82886083; E-mail: juan.iovanna@inserm.fr

is to focus on identifying pathways that are deregulated in tumors, particularly when tumor cells absolutely depend of keeping these alterations (e.g., oncogene "dependence" to survive and grow; Mancias & Kimmelman, 2011; Cohen *et al*, 2015). Consequently, it is logical to assume that blockage of these pathways with specific inhibitors should lead to cell growth arrest, death, and tumor regression. Using this rational, it would be possible to select, by means of a few markers, a particular subgroup of patients "addicted" to a distinct pathways, a major goal of modern individualized medicine.

A frequently deregulated, although insufficiently therapeutically exploited pathway in PDAC involves the "dependence" to *c-MYC* oncogene (Mertz *et al*, 2011). This transcription factor influences the expression of a significant number of genes involved in cell growth, proliferation, and apoptosis (Dang, 1999, 2012; Prendergast, 1999; Schmidt, 1999). In fact, this oncogene has been implicated in the pathogenesis of one-third of all human malignancies. As it relates to pancreatic cancer, the disease focus of the current study, *c-MYC* was found to be originally amplified in more than 30% of PDAC (Schleger *et al*, 2002) by using interphase fluorescence *in situ* hybridization, as well as overexpressed in more than 40% of tumors (Schleger *et al*, 2002). However, more recently, whole-exome sequencing of microdissected PDAC revealed that the percentage of PDAC with amplified *c-MYC* gene is approximately 12% (Witkiewicz *et al*, 2015). Early studies confirmed the oncogenic role of *c-MYC* in PDAC using genetically engineered mouse models, which upon overexpression of this gene display increased pancreatic tumorigenesis (Morton & Sansom, 2013). In addition, using a variety of experimental models, it has been later shown that upregulation of *c-MYC* is sufficient to induce the formation of PDAC without additional genetic manipulation of any cell survival pathway (Lin *et al*, 2013), deletion of one *c-MYC* allele decelerates tumor development *in vivo* (Walz *et al*, 2014), *MYC* targeted by an RNAi approach *in vivo* blocks PDAC development (Saborowski *et al*, 2014), and the subsequent increase in PGC-1α is a key determinant for the OXPHOS dependency of cancer stem cells (Sancho *et al*, 2015). Interestingly, in a more recent work, Wirth and Schneider propose to use c-MYC as a stratification marker of PDAC (Wirth & Schneider, 2016). All these features indicate that *c-MYC* behaves as a cancer driver gene for PDAC. Consequently, many efforts have been dedicated to identify potent MYC inhibitors as new therapeutic options (Soucek *et al*, 2008; Annibali *et al*, 2014; McKeown & Bradner, 2014; Fletcher & Prochownik, 2015). Key to these efforts have been the discovery that the bromodomain and extraterminal family of proteins (BET), which are efficiently inhibited by the JQ1 compound, are necessary for c-MYC activity (Nesbit *et al*, 1999; Delmore *et al*, 2011; Kandela *et al*, 2015). Notably, JQ1 suppresses PDAC development in mice by inhibiting both c-MYC activity and inflammatory signals (Mazur *et al*, 2015). Conversely, inhibition of *c-MYC* expression is thought to be also an essential mechanism by which BET inhibitors suppress tumor progression in hematological malignancies (Knoechel *et al*, 2014; Roderick *et al*, 2014; Trabucco *et al*, 2015). Thus, identifying the subgroup of pancreatic patients based on their MYC-high status and testing their response to JQ1 is timely and of paramount medical importance.

Several studies have focused on the discovery of predictive markers of response to BET inhibitors. Puissant *et al* (2013) reported that amplification of *MYCN* in medulloblastoma was the

**Table 1. Clinicopathological parameters from the learning cohort of patients.**

| | Patient distribution (learning cohort) | | |
|---|---|---|---|
| | All (%) | Resectable | Unresectable |
| *n* | 55 | 30 | 25 |
| **Sex** | | | |
| Male | 34 (62) | 18 | 16 |
| Female | 21 (38) | 12 | 9 |
| **Age** | | | |
| Mean | 64 | 66 | 61 |
| Min–Max | 41–86 | 45–86 | 41–83 |
| **Other cancers** | | | |
| No | 44 (80) | 20 | 24 |
| Yes | 11 (20) | 10 | 1 |
| **Tumor location** | | | |
| Head | 32 (58) | 19 | 13 |
| Undefined | 6 (11) | 0 | 6 |
| Body | 3 (5.5) | 2 | 1 |
| Tail | 14 (25.5) | 9 | 5 |
| **Specimen type** | | | |
| Primary tumor | 46 (84) | 30 | 16 |
| Hepatic metastasis | 5 (9) | 0 | 5 |
| Carcinomatosis | 4 (7) | 0 | 4 |
| **Tumor status at diagnosis** | | | |
| Localized | 29 (53) | 28 | 1 |
| Locally advanced | 8 (14.5) | 2 | 6 |
| Metastasis | 13 (23.5) | 0 | 13 |
| Carcinomatosis | 5 (9) | 0 | 5 |

most robust marker for predicting the sensitivity of those tumors to JQ1. Moreover, certain rare tumors called NUT midline carcinomas carrying tandem fusion of *BRD4* and *NUT* genes (nuclear protein in testis) show an important sensitivity to BET inhibitors (Stathis *et al*, 2016). However, outside to these relatively rare examples it is very difficult to predict an efficient response to the BET inhibitors by genomic approaches. To overcome this issue, the use of tumoral transcriptional program can be an effective way to develop and characterize robust predictive signatures notably in terms of chemosensitivity.

In this work, we define a transcriptomic signature that classifies PDAC, whose growth appears to depend on *c-MYC*. This result was confirmed in a prospective validation cohort of 16 independent PDAC. We determined that a third of patient-derived PDAC xenograft bear this signature and thus were likely to respond to JQ1 treatment. Indeed, experimental therapeutic studies using matching patient-derived cells in monolayer and 3D cultures confirm this prediction. We conclude that having tools to determine tumors with high c-MYC activity is of clinical interest to select patients sensitive to BET inhibitors suggesting that a similar strategy may be useful in the setting of individualized medicine efforts aimed at stratifying patients to novel treatments.

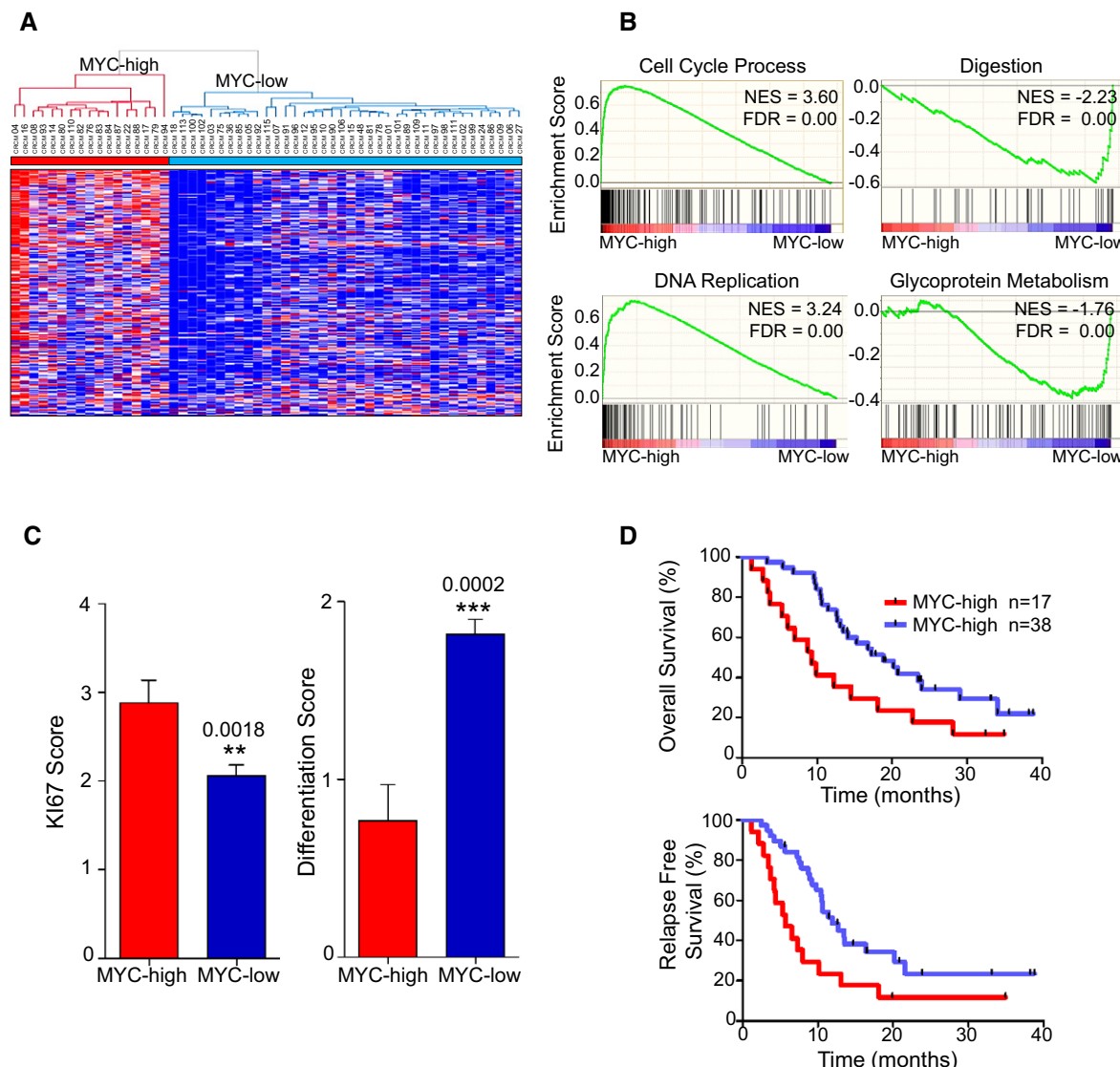

**Figure 1.   Identification of a c-MYC transcriptional signature in 55 pancreatic cancer-derived xenografts.**

A   Hierarchical clustering and expression heatmap analyzed by a non-supervised method. MYC-high and MYC-low subgroups present different expression patterns based on the selected 239 probe sets corresponding to MYC target genes (Hugene 2.0 ST Array, Affymetrix Genechips). MYC-high (*n* = 17 patients) and MYC-low (*n* = 38 patients). RMA normalized gene expression is represented in color to indicate relative gene expression (high in red, low in blue).

B   GSEA analysis of RMA normalized gene expression. Top score biological process significantly different between both groups (MYC-high and MYC-low) are represented; 825 gene sets from MSigDB collections were used. NES is the normalized enrichment score, and FDR corresponds to the false discovery rate.

C   Ki67 expression level and differentiation degree: Samples were determined by IHC and scored from 0 to 4 for Ki67 staining (0 corresponding to negative staining and 4 to maximal staining) and from 0 to 2 for differentiation state (0 corresponding to the lowest and 2 to the maximal differentiation). Pictures representing the different scores are provided in the Appendix. **$P$ = 0.0018; ***$P$ = 0.0002 (mean ± SEM, *n* = 17 vs. 38, unpaired *t*-test two-tailed).

D   Kaplan–Meier curves showing the overall (upper graph) and relapse-free survival (lower graph) for MYC-high and MYC-low subgroups. The *P*-values were calculated using log-rank test.

Source data are available online for this figure.

# Results

## Selection of PDAC patients with MYC-high or MYC-low activity by using a gene expression profile signature

In order to stratify a cohort of 55 PDAC patients, 30 primary tumors obtained from surgery and 25 biopsy samples taken by EUS-FNA were implanted subcutaneously into mice and preserved as patient-derived xenografts (PDX). The histopathologic and clinical characteristics of patients from the learning cohort are displayed in Table 1. The main anatomopathological characteristics of patient primary tumors (e.g., nuclear shape and staining intensity, nucleo-cytoplasmic ratio, eosinophilia, mucins production, and differentiation degree) were preserved in xenografts after at least six

**A**

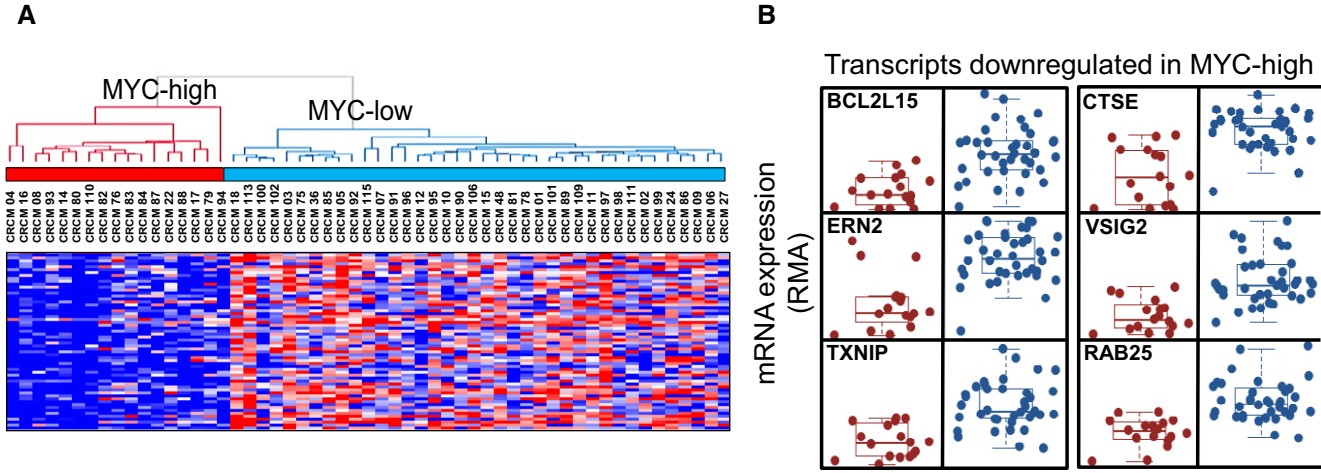

**B**

Transcripts downregulated in MYC-high

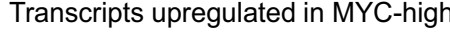

**C**

Transcripts upregulated in MYC-high

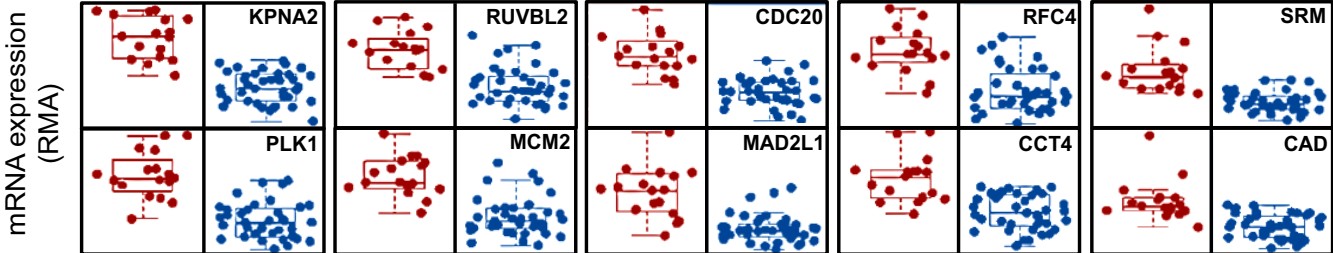

**D**

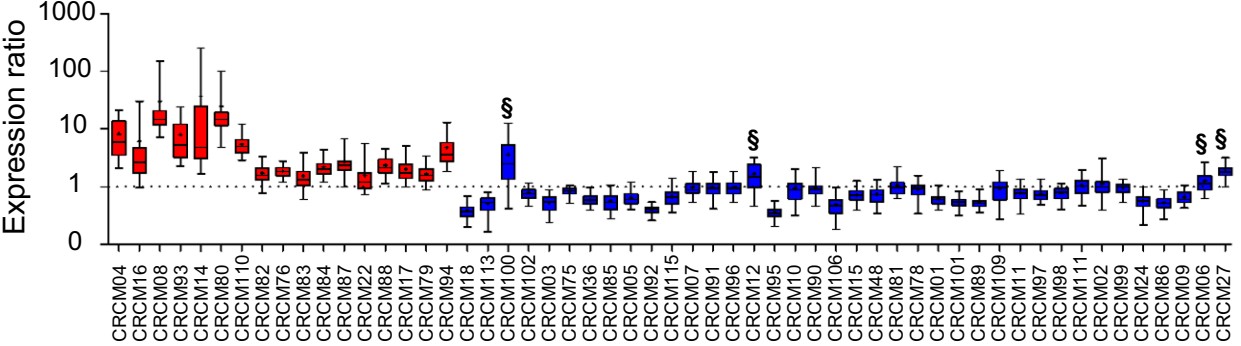

**Figure 2.  Determination of a set of 16 genes specific for MYC.**

A    Hierarchical clustering and expression heatmap for the top significantly low-expressed genes in MYC-high patients. Sixty transcripts were ranked (*P* = 0.01996; Wilcoxon *t*-test). Data correspond to RMA normalized expression values. Red and blue colors represent relative gene expression as in Fig 1A.

B, C    Box plots of the sixteen selected markers for the MYC-associated signature. In (B) are plotted the six selected transcripts that are downregulated in the MYC-high patient group, this set was selected from the 60 transcripts indicated in (A) (*P* = 0.01996 and FDR ≤ 0.044). In (C) are plotted the set of 10 MYC target transcripts upregulated in MYC-high patients which was selected from the 239 transcripts indicated in Fig 1A (*P* = 0.01996 and FDR ≤ 0.044).

D    Box plots representing the normalized expression ratios (see Materials and Methods) for the sixteen selected transcripts in the MYC-associated signature. Ratios were done with transcriptomic data obtained from the 55 patients used to select the MYC signature (training cohort). Ratios > 1 indicate a MYC-high profile, and ratios < 1 correspond to MYC-low profile. § symbols indicate the four false positives detected with the signature (duplicates [2 chips/PDX]).

Data information: The line in the box-plot representation shows the median value of mRNA expression ratios; the lower and upper limits of each box represent the first and third quartiles, respectively. Whiskers represent the limits of extreme measurements.
Source data are available online for this figure.

    

**Table 2.  List of biomarkers used in the transcriptomic signature.**

|  | Gene symbol | Affymetrix ID | RefSeq | mRNA_assignment |
|---|---|---|---|---|
| Upregulated transcripts in MYC-high cohort | CDC20 | 16663514 | NM_001255 | Homo sapiens cell division cycle 20 |
|  | KPNA2 | 16837270 | NM_002266 | Homo sapiens karyopherin alpha 2 |
|  | PLK1 | 16817017 | ENST00000300093 | Homo sapiens polo-like kinase 1 |
|  | SRM | 16681611 | NM_003132 | Homo sapiens spermidine synthase |
|  | RFC4 | 16962493 | NM_002916 | Homo sapiens replication factor C (activator 1) 4 |
|  | MCM2 | 16945101 | ENST00000265056 | Homo sapiens minichromosome maintenance complex component 2 |
|  | RUVBL2 | 16863946 | NM_006666 | Homo sapiens RuvB-like 2 (E. coli) |
|  | MAD2L1 | 16979389 | ENST00000296509 | Homo sapiens MAD2 mitotic arrest deficient-like 1 |
|  | CCT4 | 16898175 | NM_006430 | Homo sapiens chaperonin containing TCP1, subunit 4 (delta) |
|  | CAD | 16878137 | NM_004341 | Homo sapiens carbamoyl-phosphate synthetase 2 |
| Downregulated transcripts in MYC-high cohort | VSIG2 | 16745683 | NM_014312 | Homo sapiens V-set and immunoglobulin domain containing 2 |
|  | BCL2L15 | 16691121 | NM_001010922 | Homo sapiens BCL2-like 15 (BCL2L15) |
|  | RAB25 | 16671901 | NM_020387 | Homo sapiens RAB25, member RAS oncogene family |
|  | TXNIP | 16669796 | NM_006472 | Homo sapiens thioredoxin interacting protein |
|  | CTSE | 16676547 | NM_001910 | Homo sapiens cathepsin E, transcript variant 1 |
|  | ERN2 | 16825120 | NM_033266 | Homo sapiens endoplasmic reticulum to nucleus signaling 2 |

successive passages (Duconseil *et al*, 2015). Growth rates to reach a tumor volume of 1 cm$^3$ ranged from 2 to 6 months in most of the PDX. Total RNA was obtained from the 55 PDX, and gene expression profiling was performed using Affymetrix platform. Subsequently, we selected a panel of 239 RNAs regulated by c-MYC in accordance with the MYC targets v1 and v2 list from Molecular Signatures Database (MSigDB). Figure 1A represents the hierarchical clustering and heatmap for the top significantly high-expressed genes in MYC-high patients. The dendrogram showing the genetic distance between patients indicates the presence of two major subgroups that we define as MYC-high and MYC-low in red and blue colors, respectively. Interestingly, we observe that 17/55 (30.9%) patients are characterized by an increase in the expression of 134/239 c-MYC target RNAs (*P*-value = 0.01996 and *q*-value (FDR) = 0.044). The rank-listed transcripts are available in Appendix Table S1. In order to gain insight into the potential biological processes enriched in MYC-high vs. MYC-low subgroups, we performed a Gene Set Enrichment Analysis (GSEA). As shown in Fig 1B, the MYC-high subgroup is characterized by a low differentiated phenotype and their two most significant associated biological processes are cell cycle process (Normalized Enrichment Score = 3.60 and FDR = 0.00) and DNA replication and genome maintenance (Normalized Enrichment Score = 3.24 and FDR = 0.00). In contrast, the MYC-low subgroup is characterized by biological processes that reflect a more differentiated state of pancreatic tumors such as

digestion (Normalized Enrichment Score = −2.23 and FDR = 0.00) and glycoprotein metabolism (Normalized Enrichment Score = −1.76 and FDR = 0.00). In addition, a complete list of statistically significant enriched signatures using Biological Process, Curated Geneset Enriched, and Hallmarks Enriched tools is presented in Datasets EV1–EV6. To confirm that MYC-high patients give rise to PDX with high proliferative index, we performed an IHC-based Ki67 staining scoring on the epithelial compartment of the 55 PDX. As shown in Fig 1C (left part), this semi-quantitative scoring reveals that MYC-high patient-derived PDX proliferate more than the MYC-low subgroup (Ki67 mean score 2.88 ± 0.25 [*n* = 17] vs. 2.06 ± 0.12 [*n* = 38], *P* = 0.0018). In addition, we determine the degree of differentiation for both subgroups on H&E staining on paraffin-embedded tissues sections. As shown in Fig 1C (right part), MYC-high subgroup shows lower differentiation state than MYC-low subgroup (differentiation mean score 0.77 ± 0.2 [*n* = 17] vs. 1.82 ± 0.08 [*n* = 38], *P* = 0.0002). The Ki67 and differentiation scores are provided in Appendix Fig S1A and B, respectively. Moreover, we analyzed the clinical outcome of both MYC-high and MYC-low patients using a Kaplan–Meier analysis and considering both the overall and the relapsing free survival time for the 55 patient cohort. As shown in Fig 1D, the overall survival median is 9.2 months for the MYC-high vs. 18.8 months for the MYC-low subgroup (HR = 2.43 [1.1–5.1]). The relapse-free survival median is 5.6 and 11.5 months for MYC-high and MYC-low subgroup, respectively (HR = 2.7 [1.3–

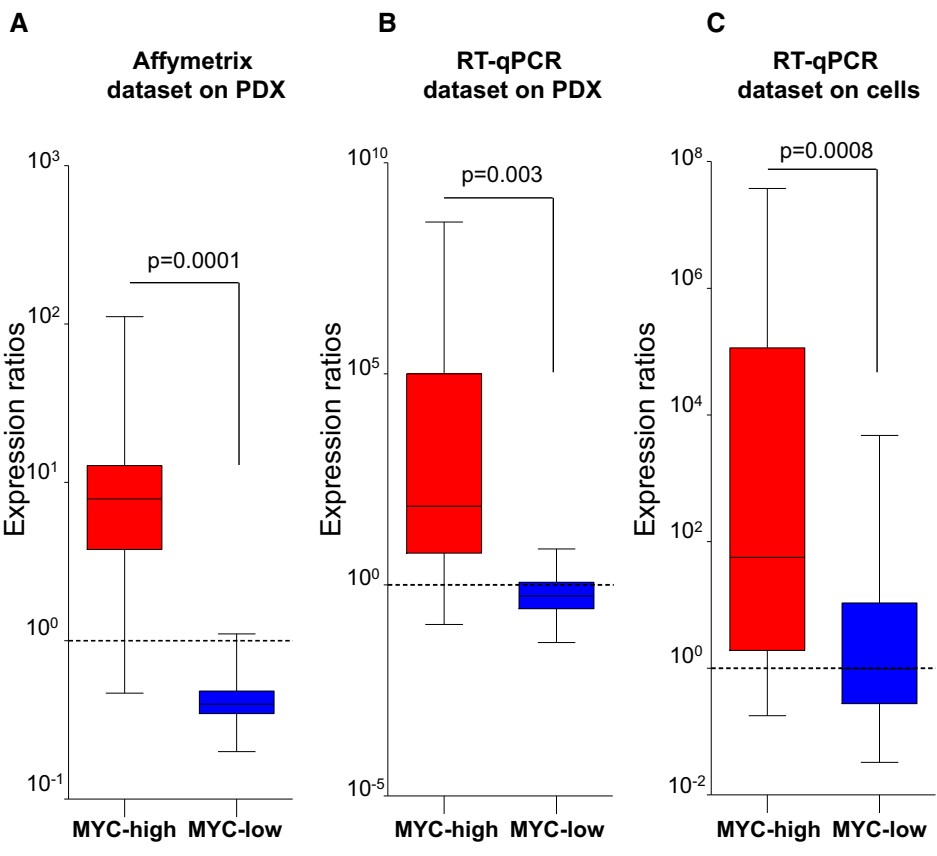

**Figure 3.  Validation of the MYC signature.**

A–C  Box plots representing normalized expression ratios (A) from transcriptomic data as in Fig 2D for four PDX of each group (MYC-high and MYC-low), (B) by RT–qPCR for the same PDX as in (A), and (C) by RT–qPCR for the same eight PDX-derived primary cell cultures. The dotted line indicates the threshold 1 that differentiates between MYC-high and MYC-low profiles. The line in the box-plot representation shows the median value of mRNA expression ratios, the lower and upper limits of each box represents the first and third quartiles, respectively. Whiskers represent the limits of extreme measurements (Mann–Whitney *t*-test).

Source data are available online for this figure.

5.8]). Altogether, these observations indicate that we can identify patients with MYC-high and MYC-low activity. Moreover, PDAC with MYC-high activity is characterized by increased proliferation, lower differentiation status and they have poor survival expectancy. Combined, these observations constitute a solid characterization of the molecular, biological, and medical features of the *c-MYC* status in patient-derived xenografts, which is necessary to build the trajectory toward the testing of novel therapies aimed at treating this distinct subgroup of tumors.

**MYC-dependent RNA signatures can be used for classifying distinct PDAC subtypes**

To define a specific MYC signature that can be used to classify tumor subtypes, we selected a total of 16 genes. The first 10 (Figs 1A and 2C) were identified from the gene set corresponding to the upregulated genes in the MYC-high group of patients. To obtain the genes downregulated in the MYC-high subgroup, we identified the six top-score downregulated genes in the MYC-high patients by a *t*-test analysis from the whole gene expression profiles (Fig 2A

and B). The list of these 16 markers is shown in the Table 2. For each patient, 60 ratios were computed after mean centered normalization of the 16 markers (see Materials and Methods for the normalization method) revealing the MYC-high or MYC-low profiles. As shown in Fig 2D, we were able to detect the 17 MYC-high patients with medians of expression ratios up to 1 with an excellent specificity and accuracy. Affymetrix data were then confirmed by RT–qPCR on four putative MYC-high and four MYC-low patients. Each transcript was normalized to the 28S ribosomal RNA, the relative quantity was calculated by the $\Delta\Delta C_t$ method and the ratios were calculated after normalization. The signature was able to definitively detect all MYC-high and MYC-low profiles by RT–qPCR as shown in Fig 3A and B. We then assessed the MYC signature on primary cultures derived from the same eight xenografts (Fig 3C) and found that the corresponding MYC-high or MYC-low profiles were correctly detected. Therefore, we conclude that the signature based on these transcripts is a reliable for identifying tumor subtypes based on their *c-MYC* status.

We also analyzed whether or not the genetic alterations of the *c-MYC* gene in PDAC tumors are predictive of the response to BET

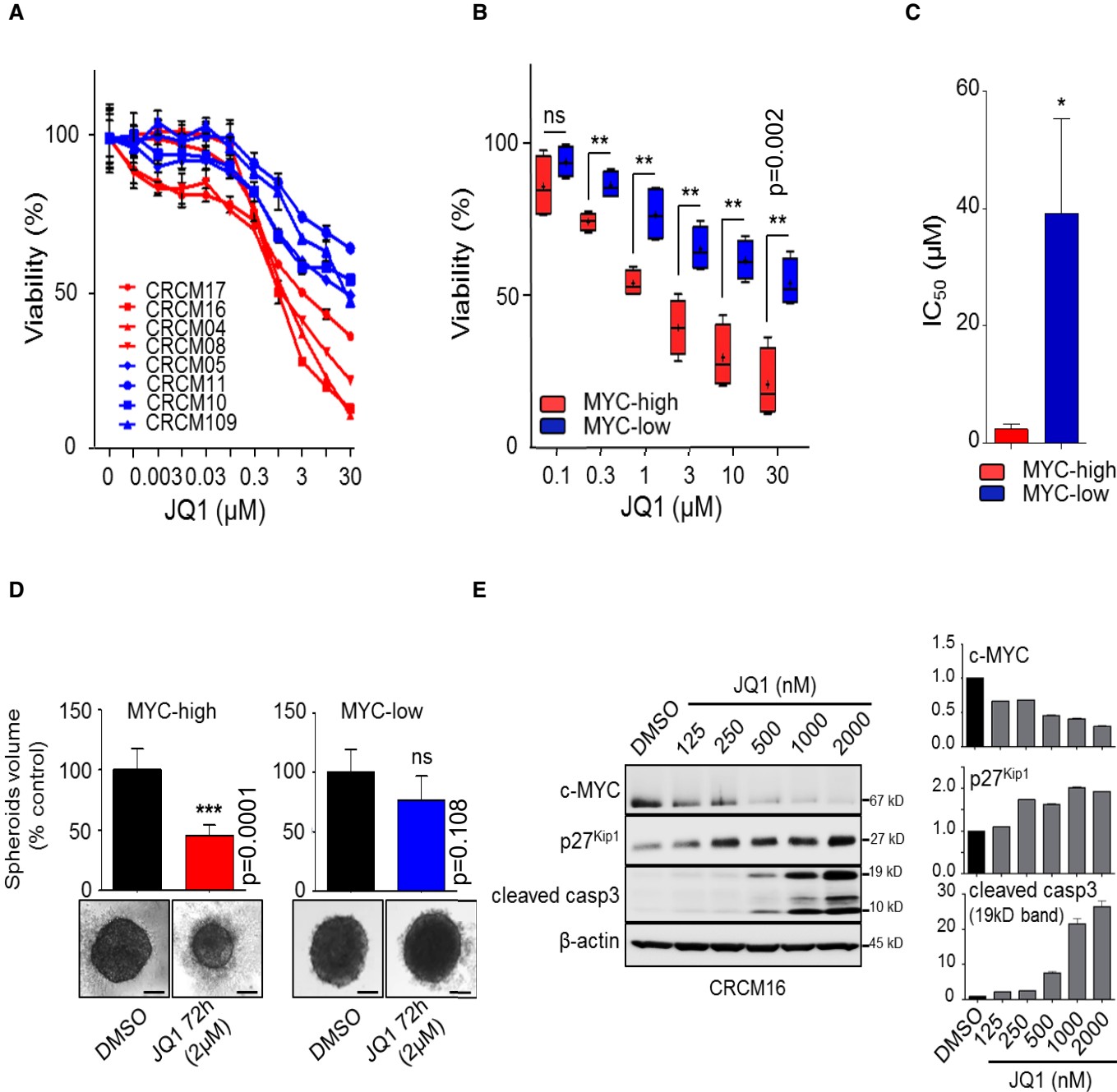

**Figure 4.  MYC-high PDX were sensitive to the JQ1 compounds.**

A  Chemograms for eight PDX-derived cell lines. Dose–response curves after 72 h of JQ1 treatment. Cell viability is indicated in % to the control (vehicle treated). Error bars represent SEM; $n$ = 3.

B  Box plots representing JQ1 sensitivity for the six highest concentrations used in chemograms. The line in the box-plot representation shows the median value of mRNA expression ratios; the lower and upper limits of each box represent the first and third quartiles, respectively. Whiskers represent the limits of extreme measurements (**$P$ = 0.002, unpaired $t$-test with Welch's correction, triplicates).

C  Histograms representing $IC_{50}$ for JQ1 for the four MYC-high and the four MYC-low cell lines (*$P$ = 0.05, unpaired $t$-test with Welch's correction).

D  Histograms representing spheroid volumes from three derived cell lines in each group treated with 2 μM JQ1 for 72 h or with DMSO (0.05%). Data are representative of three independent experiments made in triplicate. Phase contrast pictures were taken under a 4×/NA 0.45 μm objective lens. Black scale bars represent 100 μm (***$P$ = 0.0001, unpaired $t$-test with Welch's correction).

E  Expression of c-MYC, p27kip1 and cleaved caspase-3 of CRCM16 in primary cells treated with increasing concentrations of JQ1 or vehicle (DMSO). The graphs represent the densitometry of each protein normalized on the β-actin levels. Densitometries were made with ImageJ software (ImageJ, National Institutes of Health, Bethesda, Maryland, USA) (duplicates).

Source data are available online for this figure.

**Table 3. Clinicopathological parameters from the validation cohort of patients.**

| | Patient distribution (validation cohort) | | |
| --- | --- | --- | --- |
| | All (%) | Resectable | Unresectable |
| *n* | 16 | 7 | 9 |
| Sex | | | |
| Male | 10 (62.5) | 3 | 7 |
| Female | 6 (37.5) | 4 | 2 |
| Age | | | |
| Mean | 69 | 67 | 70 |
| Min–Max | 53–83 | 58–71 | 53–83 |
| Other cancers | | | |
| No | 11 (68.75) | 4 | 7 |
| Yes | 5 (31.25) | 3 | 2 |
| Tumor location | | | |
| Head | 5 (31.25) | 3 | 2 |
| Undefined | 2 (12.5) | 0 | 2 |
| Body | 2 (12.5) | 1 | 1 |
| Tail | 7 (43.75) | 3 | 4 |
| Specimen type | | | |
| Primary tumor | 13 (81.25) | 7 | 6 |
| Hepatic metastasis | 2 (12.5) | 0 | 2 |
| Carcinomatosis | 1 (6.25) | 0 | 1 |
| Tumor status at diagnosis | | | |
| Localized | 7 (43.75) | 7 | 0 |
| Locally advanced | 1 (6.25) | 0 | 1 |
| Metastasis | 6 (37.5) | 0 | 6 |
| Carcinomatosis | 2 (12.5) | 0 | 2 |

inhibitors. To this end, we examined the gain (one or more alleles) of *c-MYC* gene in the PDX collection. We provide the *c-MYC* gain status for both MYC-high and MYC-low samples in Appendix Fig S2. This analysis reveals that 15 of 17 PDX samples with a MYC-high phenotype show a gain in the c-Myc gene copy number, but also 20 of 38 PDX samples from the MYC-low group. Thus, in the MYC-low samples, which are not good responders to JQ1, approximately half of patients also display increases in *c-MYC* copy number. This phenomenon suggests that potential epigenetic mechanisms are deployed by cells to compensate for the increase in *c-MYC* copy number in this tumor. Based on this observation, we conclude that *c-MYC* CNV alterations, although more frequent in MYC-high tumors, are an unsuitable prediction method to estimate the higher BET inhibitors sensitivity.

### MYC-high PDX are sensitive to growth inhibition by the BET inhibitor JQ1 *in vitro*

We hypothesized that the subgroup of PDX belonging to the MYC-high phenotype should be more sensitives to pharmacological inhibition of MYC activity, which currently cannot be targeted directly but instead through the inactivation of BET proteins. To test our

hypothesis, we treated a panel of pancreatic PDX-derived primary cultures with the well-characterized BET inhibitor JQ1. According to their MYC signature, we selected four MYC-high PDX (CRCM16, CRCM17, CRCM04, and CRCM08) and four MYC-low patients (CRCM05, CRCM11, CRCM10, and CRCM109) (see Fig 2D). We assessed the viability of cells with increasing dose of drug (chemograms) for 72 h. As shown in Fig 4A and B, MYC-high cells exhibit higher sensitivity to JQ1 treatment compared to the MYC-low ones. The mean of $IC_{50}$ for the MYC-high cells is 2.3 μM ± 0.8, whereas the one corresponding to MYC-low primary cells was of 39.22 μM ± 16 (Fig 4C). We also evaluated the effect of JQ1 in patient-derived cells grown in 3D culture conditions. As shown in Fig 4D, MYC-high spheroids were more sensitive to the JQ1 treatment for 72 h (50% reduction in volume) than their MYC-low counterpart (25% reduction in volume). As a positive control, we analyzed the effect of JQ1 treatment on the MYC-high primary cell CRCM16 and found a significant decrease of MYC protein level (Fig 4E). Importantly, MYC depletion is accompanied with an increase in p27$^{Kip1}$ level and an increase in the cleavage of caspase-3 which may explain the antitumor effect of the compound. Thus, these experimental therapeutic experiments suggest that inhibition of BET proteins by small drugs like JQ will be beneficial to antagonize the growth of pancreatic cells which carry the MYC-high status.

### MYC-dependent RNA signature identify MYC-high patients on an independent validation cohort

Sixteen new PDAC patients were included in the study as an independent validation cohort. The histopathologic and clinical characteristics of patients from the test cohort are displayed in Table 3. We obtained 16 PDX and from them six PDX-derived cells. We measured the expression of 16 MYC-associated markers by RT–qPCR and found that eight patients present a MYC-high profile (CRCM43, CRCM26, CRCM50, CRCM19, CRCM30, CRCM114, CRCM116, and CRCM34) and eight show a MYC-low profile (CRCM23, CRCM21, CRCM108, CRCM25, CRCM28, CRCM112, CRCM29, CRCM42) as described in Fig 5A. Of the six primary cultures available, three presented a MYC-high (CRCM116, CRCM114, and CRCM34) and three a MYC-low profile (CRCM112, CRCM21, and CRCM28). To test their sensitivity to BET inhibitors, cells were treated with increasing concentrations of JQ1 and as expected the three MYC-high cell cultures showed to be more sensitive than the MYC-low cells as shown in Fig 5B and C. The mean of $IC_{50}$ for the MYC-high cells is 5.65 μM ± 2.4, whereas it is 223.71 μM ± 191.5 for the MYC-low primary cultures (Fig 5D) which are close to the study cohort presented in Fig 5C. These results confirm that cells from PDAC presenting a MYC-high profile are more sensitive to JQ1 treatment compared to the cells presenting a MYC-low profile.

### MYC-high PDX are sensitive to the BET inhibitor JQ1 in nude mice model

Finally, we performed a preclinical analysis by treating four PDX with MYC-high and four with MYC-low phenotypes with the JQ1 compound (50 mg/kg/day) to validate the obtained *in vitro* results. As shown in Fig 6, CRCM16, CRCM04, CRCM114, and

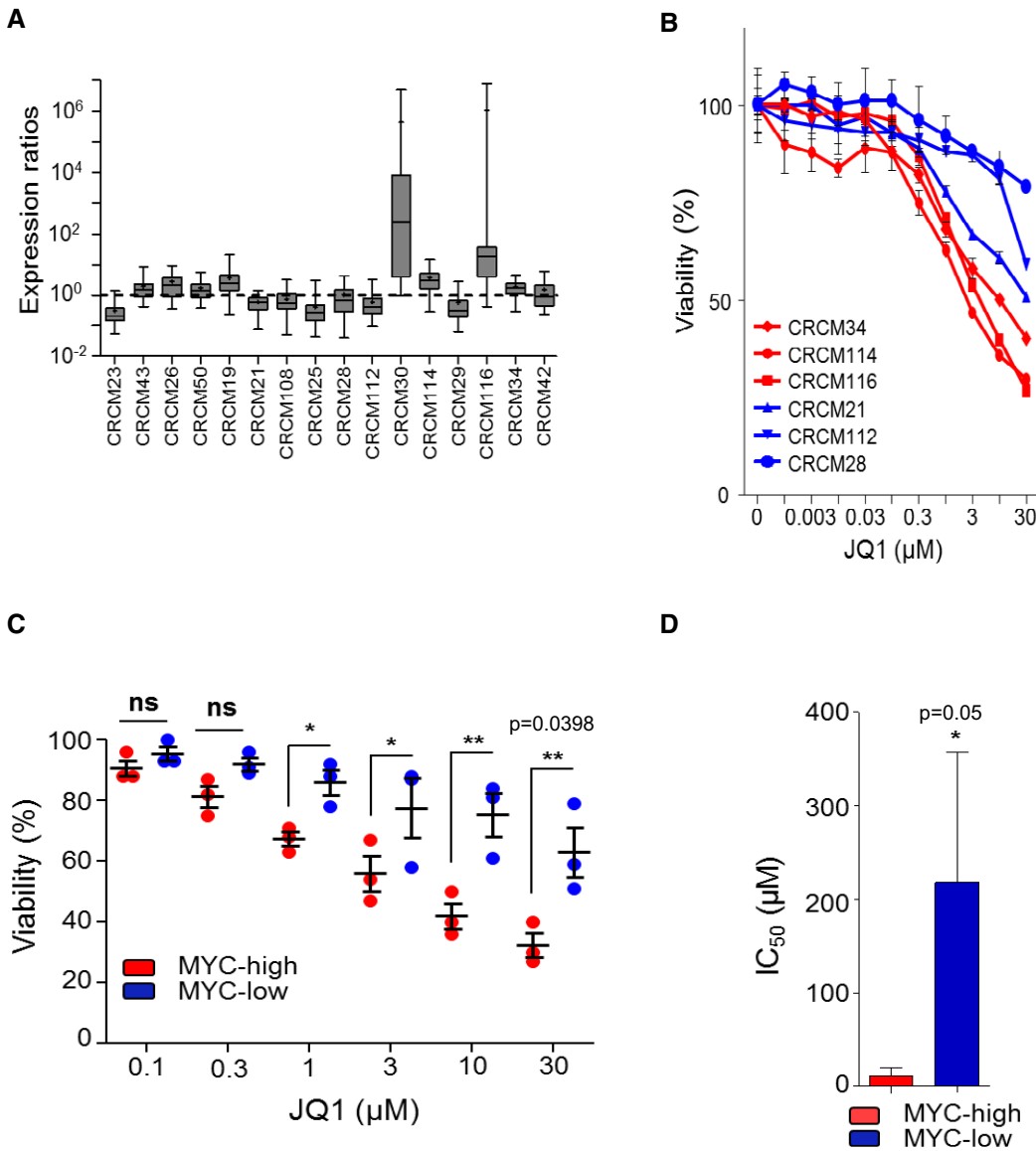

**Figure 5. Verification in a validation cohort.**

A Box plots representing normalized expression ratios for the MYC signature in 16 new PDX used as validation or test cohort. The line in the box-plot representation shows the median value of mRNA expression ratios; the lower and upper limits of each box represent the first and third quartiles, respectively. Whiskers represent the limits of extreme measurements. qPCR in duplicate for each PDX.

B Chemograms for eight PDX-derived cell lines with MYC-high (red) or MYC-low (blue) profiles were subjected to JQ1 treatment as in Fig 4A. Cell viability is indicated as % of the control (vehicle treated). Error bars represent SEM; $n = 3$.

C Graph representing JQ1 sensitivity for the six highest concentrations used in chemograms. Horizontal lines represent the median $\pm$ SEM (*$P = 0.014$; **$P = 0.0398$, Welch's $t$-test).

D Histograms representing $IC_{50}$ for JQ1 for the three MYC-high and the three MYC-low cell lines taken from the validation cohort (mean $\pm$ SEM, unpaired $t$-test with Welch's correction).

Source data are available online for this figure.

CRCM116 (MYC-high) samples efficiently responded to treatment. On the contrary, samples with the MYC-low phenotype (CRCM05, CRCM10, CRCM109, and CRCM112) were more resistant. Altogether, from the *in vitro* and *in vivo* results, we can assume that MYC-high tumors are more sensitive to the BET inhibitors.

# Discussion

Current treatments for patients with a PDAC are not highly effective primarily due to the recently discovered fact that these tumors are both molecularly and clinically heterogeneous. For example, the response of these tumors to gemcitabine and Folfirinox, the two

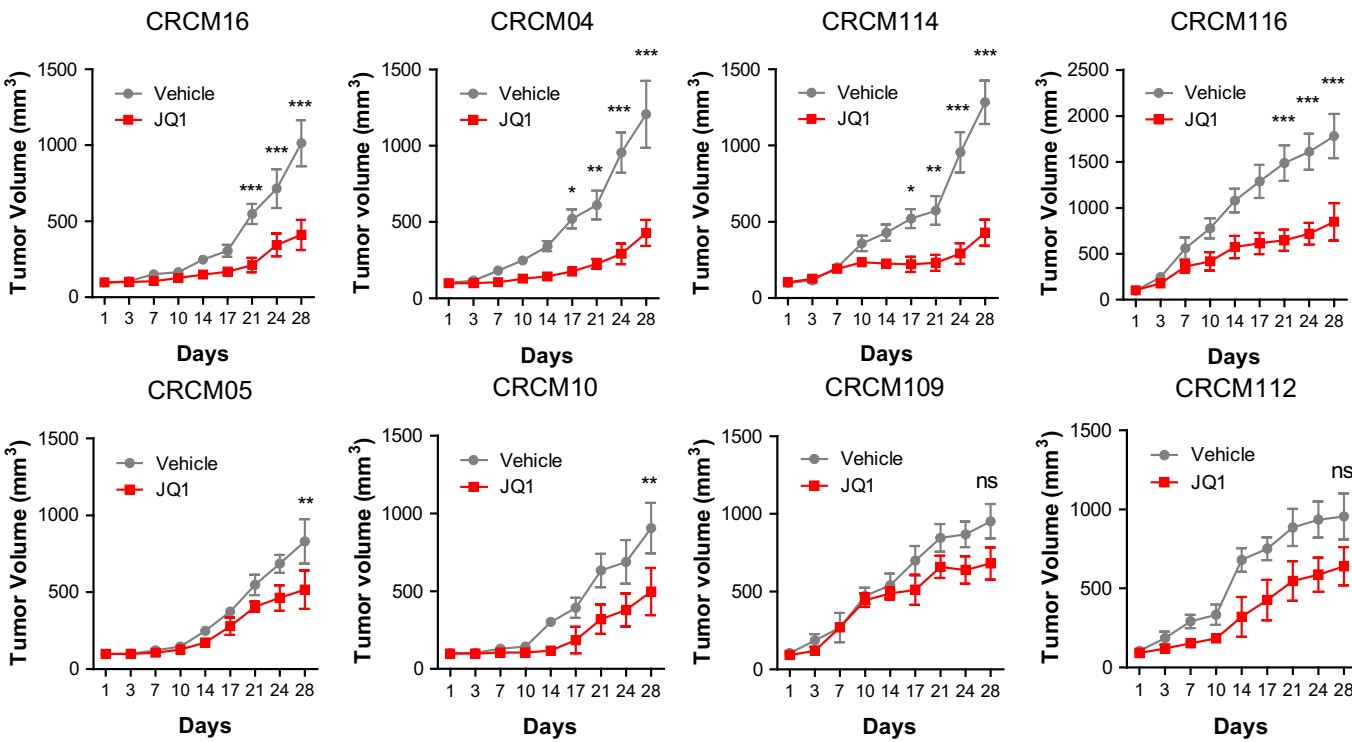

**Figure 6. JQ1 is a more potent inhibitor in MYC-high PDX *in vivo*.**
Four PDX with MYC-high (CRCM16, CRCM04, CRCM114, and CRCM116) and four with MYC-low (CRCM05, CRCM10, CRCM109, and CRCM112) phenotypes were treated with 50 mg/kg/day of JQ1 or with vehicle for 28 days by i.p. injections ($n$ = 3 per group). ***$P$ < 0.001, **$P$ < 0.01 and *$P$ < 0.05 in two-way ANOVA with Bonferroni post-test (mean ± SEM).
Source data are available online for this figure.

gold standard therapies against PDAC, is only 10% (Burris *et al*, 1997) and 31% (Conroy *et al*, 2011), respectively. The variability in this response seems to be due, on one hand, to the difficulty for the drugs to reach the transformed cells because of the compact PDAC stroma (resulting in hypovascularization) and, on the other hand, to the marked differences in cellular susceptibility to drugs into the tumors due to their molecular heterogeneity. Therefore, it has become important to develop methods to stratify patients in a manner that allows predicting their susceptibility to the treatments so as to increase their therapeutic responses which will result in increased survival expectancies. Consequently, in this work, we focused our attention on a subgroup of PDAC, which are characterized by a deregulation of the *c-MYC* pathway, established a molecular signature that allows us to identify those tumors of the MYC-high subtype and showed that they are highly sensitive to BET inhibitors. Based on this observation, we conclude that the development of tools to determine tumors with high deregulation in the *c-MYC* pathway is of clinical interest to select patients sensitive to BET inhibitors, a goal that can help with the design and administration of individualized medicine protocols.

The oncogene *c-MYC* is the "corner stone" of several proliferative pathways involved in PDAC development such as RAS-RAF-MEK-ERK and PI3K-AKT pathways that stabilize c-MYC by preventing its degradation by the proteasome (Sears *et al*, 2000; Dai *et al*, 2006; Zhu *et al*, 2008). Consistent with these observations, we chose to define c-MYC transcriptional program as a restricted signature that

allowed the selection of distinct subtype of PDAC tumors. This MYC-associated signature was established by using selected c-MYC target genes available from the molecular signature database and who are reported to be directly regulated by this transcription factor (MYC target v1 and v2 from MsigDB). From the 239 putative c-MYC-dependent genes, 16 targets were selected according to the fold change between MYC-high and MYC-low PDAC. Those targets usefully identify highly proliferative PDAC with low degree of differentiation and patients with poor clinical outcome as shown in Fig 1C and D. An efficient signature, like the one reported here, which is easily applicable and low cost, for detecting patients having a MYC-high PDAC is of clear clinical interest, particularly in non-operable patients which represent about 85% of PDAC. Currently, in these patients, a biopsy is systematically taken by EUS-FNA before starting the antitumor treatment as a diagnosis confirmation procedure. These biopsies represent a valuable source of cancer cells which may serve as the source of tumor RNA. In turn, this RNA may be used for measuring the expression of a set of RNAs of interest. We envision that in the near future treatments of cancer patients will be preceded by the molecular characterization of their tumor in order to select the more appropriate treatments toward an individualized medicine approach. PDAC is undoubtedly one of the malignant diseases that most urgently need this type of approaches since the treatment with standard available drugs are almost inefficient. Conceptually, targeting on the pathways, which are dominantly deregulated such

as those driven by oncogenes such as Kras and *c-MYC* by specific drugs, could be of major interest for patients and we presume that it may have repercussions in terms of survival. This work was focused on *c-MYC*, but a similar approach can be applied to other pathways for PDAC as well as for other cancer types. Our work was also benefited by the existence of well-characterized drugs that reliably inhibit the *c-MYC* pathway such as JQ1, a thienotria-zolodiazepine, and a potent inhibitor of the BET family of proteins which includes BRD2, BRD3, BRD4, and BRDT (Filippakopoulos *et al*, 2010; Asangani *et al*, 2014). Mechanistically, JQ1 competitively inhibits the BET proteins from binding to acetylated lysine residues of histones (Jung *et al*, 2014; Korb *et al*, 2015). This process prevents the association of transcriptional complexes with chromatin and thus decreases expression of RNA species that are dependent of this mechanism of transcription (Smith & Zhou, 2016). Many studies suggest that the main mechanism by which BET inhibitors affect tumoral growth is by their effects on *c-MYC* expression and activity (Knoechel *et al*, 2014; Roderick *et al*, 2014; Trabucco *et al*, 2015). However, a recent paper suggests an additional antitumor effect of JQ1 on PDAC via the inhibition of CDC25B, a regulator of cell cycle progression. Whether this effect is specific to JQ1 or common to BET inhibitors remains to be demonstrated (Garcia *et al*, 2016). Lastly, in hepatocellular carcinoma, MYC expression does not seem to be predictive of the response of these tumors to JQ1 (Huang *et al*, 2014). Therefore, we can assume that MYC level is not indicative of the MYC activity or, alternatively, that the response to JQ1 could be specific in certain tissues, although testing both of these hypotheses remains the topic of future investigations.

In conclusion, this study is the first to report a strategy, which begun by molecularly characterizing patient-derived PDX as well as to define a molecular signature that can help to select PDAC patients with deregulation in the c-MYC pathway and showing that these tumors are more sensitive to the BET inhibitor treatment. We presume that a therapeutic strategy against PDAC by using BET inhibitors, in combination with standard anticancer drugs, could be a promising strategy in well-selected patients. A similar approach can be applied to other pathways for PDAC or other cancer types, highlighting its potential to contribute to the development of novel individualized therapies for malignant diseases, which currently remain incurable.

# Materials and Methods

### Ethics statements

The study was performed according to the principles set out in the WMA Declaration of Helsinki and to the protocols approved by the Department of Health and Human Services Belmont Report. Patients were included in this project under the Paoli Calmettes Institute clinical trial NCT01692873 (https://clinicaltrials.gov/show/NCT01692873). Three expert clinical centers collaborated on this project after receiving ethics review board approval. Patient informed consent forms were collected and registered in a central database.

For animal studies, all mice, housed under specific pathogen-free conditions, were 4-week-old males on a Swiss nude

background (Crl: Nu(lco)-Foxn1$^{nu}$, Charles River, Wilmington, MA). Mice were used at 6 weeks of age, and all animal studies were approved by the Animal Facility and Experimental Platform (Scientific and Technological Park of Luminy, Marseille, France) and the French Ministry of National Education and Research under the reference number: 02859.01. All animal experiments were conducted in accordance with the Guides for the Use and Care of Laboratory Animals (ARRIVE guidelines). The animal house is run by professional employees fully equipped with state-of-the-art instrumentation in order to maintain the standard of animal welfare at the maximum levels. All mice were housed in individual, ventilated cages (IVCs) with 12-h light/dark cycles with food and water *ad libitum*.

### PDAC samples and cell culture

Two types of samples were obtained: endoscopic ultrasound-guided fine-needle aspiration (EUS-FNA) biopsies from patients with unresectable tumors and tumor tissue samples from patients undergoing surgery. All the samples were anonymized, and postsurgical anatomopathology reports were provided. Histopathologic evaluation was performed on 5-μm hematoxylin and eosin stained sections of patient tumors and xenografts and examined under a light microscope. These sections were compared with the original human tumor when available. Samples from EUS-FNA were mixed with 100 μl of Matrigel™ (BD Biosciences, Franklin Lakes, NJ) and injected in the upper right flank of a nude mouse (Swiss Nude Mouse Crl: NU(lco)-Foxn1nu; Charles River Laboratories, Wilmington, MA). Samples from surgery were fragmented, mixed with 100 μl of Matrigel™, and implanted with a trocar (10 gauge; Innovative Research of America, Sarasota, FL) in the subcutaneous right upper flank of an anesthetized and disinfected mouse. When the tumors reached 1 cm$^3$, the mice were sacrificed, and the tumors were removed. Xenografts that failed to develop within 6 months were discontinued. The study on animals was approved by the Animal Facility and Experimental Platform (Scientific and Technological Park of Luminy, Marseille, France).

Primary cell cultures were obtained from xenografts. Tissues were split into several small pieces and processed in a biosafety chamber. After a fine mincing, they were treated with collagenase type V (ref C9263; Sigma) and trypsin/EDTA (ref 25200-056; Gibco, Life Technologies) and suspended in DMEM supplemented with 1% w/w penicillin/streptomycin (Gibco, Life Technologies) and 10% fetal bovine serum (Lonza). After centrifugation, cells were resuspended in Serum Free Ductal Media (SFDM) adapted from Schreiber *et al* (2004) without antibiotic and incubated at 37°C in a 5% CO$_2$ incubator.

### Gene expression microarrays

Total RNA was purified from xenograft using TRIzol® Reagent (Gibco, Life Technologies) according to the manufacturer. Briefly, 50–100 mg of fresh frozen tissue per ml of TRIzol® was disrupted using a homogenizer followed by a single step of phenol/chloroform purification. Total RNA was quantified using the Nanodrop spectrophotometer (NanoDrop Technologies, Inc), and RNA Integrity Number (RIN) was calculated using the Agilent 2100 Bioanalyzer (Agilent Technologies, Santa Clara, CA). RNA samples that

reached a RIN between 8 and 10 were used for microarray hybridization (GeneChip; Affymetrix Inc., Santa Clara, CA). The Genechip® Human Gene 2.0 ST Arrays were washed and stained using the Affymetrix GeneChip fluidic station 450 (protocol EukGE-WS2v5_450) and were scanned using a GeneChip scanner 3000G7 (Affymetrix Inc., Santa Clara, CA). GeneChip operating software version 1.4 (Affymetrix Inc., Santa Clara, CA) was used to obtain chip images and for quality control. Background subtraction and normalization of probe set intensities were performed using the method of Robust Multi-array Analysis (RMA; Irizarry *et al*, 2003). Microarray analysis was performed by the CHU de Québec Research Center Gene Expression Platform (Quebec City, Quebec, Canada). Seventeen samples of the cohort were previously published (Duconseil *et al*, 2015) as GEO accession numbers GSE55513 and GSE89792.

## Bioinformatics analysis

Robust Multi-array Analysis normalized data from microarrays were imported into GENE-E (version 3.0.204; Broad Institute, Cambridge, MA, USA) to generate heatmaps. The color intensity on the heatmap reflects global expression within a minimum (25%) in blue and a maximum (75%) in red. Cluster analysis, using Euclidean distance correlation of samples only, and distance for the clustering were calculated using a complete linkage algorithm. Differentially expressed genes were identified using *t*-test ratio, and false discovery rate was estimated using Benjamini & Hochberg method (Benjamini *et al*, 2001). Gene set enrichment analysis (GSEA) was performed using the Broad Institute platform, and statistical significance (false discovery rate) was seated at 0.05.

## GSEA analysis

Two categories of pre-defined gene sets in the Molecular Signatures Database (MSigDB, Broad Institute, Cambridge, MA, USA) were selected for analysis named the Hallmarks set, and the C5 set, a Gene Ontology (GO) molecular function gene set derived from the Molecular Function Ontology database (Subramanian *et al*, 2005). The gene sets included in the analysis were limited to those that contained between 15 and 500 genes. Permutation was conducted 1,000 times according to default-weighted enrichment statistics and by using a *t*-test ratio metric to rank genes according to their differential expression levels across the MYC-high and MYC-low groups. Significant gene sets were defined as those with a nominal $P < 0.05$. Calculation of the false discovery rate (FDR) was used to correct for multiple comparisons and gene set sizes (Benjamini *et al*, 2001).

## SNP arrays analysis

DNA was extracted for 55 PDX samples using the Blood & Cell culture DNA mini kit (Qiagen) following the manufacturer's instructions. Illumina Infinium HumanCode-24 BeadChip SNP arrays were used to analyze the DNA samples. Integragen SA (Evry, France) carried out hybridization, according to the manufacturer's recommendations. The BeadStudio software (Illumina) was used to normalize raw fluorescent signals and to obtain log R ratio (LRR) and B allele frequency (BAF) values. Asymmetry in BAF signals

due to bias between the two dyes used in Illumina assays was corrected using the tQN normalization procedure (Staaf *et al*, 2008). We used the circular binary segmentation algorithm (Venkatraman & Olshen, 2007) to segment genomic profiles and assign corresponding smoothed values of log R ratio and B allele frequency. The Genome Alteration Print (GAP) method was used to determine the ploidy of each sample, the level of contamination with normal cells, and the allele-specific copy number of each segment (Popova *et al*, 2009). Chromosomal instability index (CIN) was estimated by the mean number of SNP probes with a loss or gained status normalized by chromosomes length. SNP data are available through ArrayExpress (http://www.ebi.ac.uk/arrayexpress) under accession E-MTAB-5006.

## Chemograms

Cells were screened for their chemosensitivity to JQ1 compound (Sigma-Aldrich, St Louis, MO, USA). Five thousand cells per well were plated in 96-well plates in SFDM medium. Twenty-four hours later, the media was supplemented with increasing concentrations of JQ1 and incubated for an additional 72-h period. Each experiment was done in triplicate and repeated at least three times. Ten increasing concentrations of JQ1 were used ranging from 0 to 30 μM.

## Spheroids outgrowth assay

Fifteen thousand cells per well were seeded in 96-well round bottom plates with medium containing 20% methylcellulose (Sigma-Aldrich, St Louis, MO, USA). After 48-h incubation, cells with spheroids of uniform size and shape were incubated with 2 μM JQ1 during 72 h. Images were captured every day with an Evos microscope, equipped with a 4×/N.A. (0.13) objective lens (Thermo Fisher Scientifics, Waltham, MA, USA). Spheroids volumes were determined by the following equation: $V_{spheroids} = 4/3 \ \pi{*}r^3$. Results were expressed as a percentage of spheroid growth compared with no treatment condition (DMSO 0.05%).

## Viability assays

Cell viability was estimated after addition of PrestoBlue™ reagent (Life Technologies, Carlsbad, CA, USA) for 3 h, following the supplier protocol. For spheroid viability assays, the viability was estimated after addition of CellTiter-Glo® 3D for 1 h, following the supplier protocol (Promega, Madison, USA).

## Proteins extraction and Western blotting

Cells were washed with ice-cold PBS and lysed in Laemmli sodium dodecyl sulfate-sample buffer (90 mM Tris–HCl [pH 6.8], 2.5% sodium dodecyl sulfate, 15% glycerol). Samples were then boiled and sonicated, and protein concentrations were determined using the bicinchoninic acid (BCA) assay (Bio-Rad, Hercules, CA, USA) with bovine serum albumin as standard. β-mercaptoethanol and bromophenol blue were then added to a final concentration of 1% and 0.005%, respectively. Proteins (20 μg) were separated by SDS–PAGE in 10% or 12.5% gels and were detected immunologically following electro-transfer onto equilibrated PVDF (Imobilon-P

membranes, Millipore, Billerica, MA, USA). PVDF membranes were stained with Ponceau Red to assure a correct transfer of proteins and molecular weight markers. Membranes were blocked in PBS containing 5% powdered milk and 0.05% Tween-20 for 1 h at 25°C. Membranes were then incubated overnight at 4°C with primary antibodies in blocking solution and thereafter with horseradish peroxidase-conjugated IgG for 1 h. Blots were visualized using the Amersham ECL system. The c-MYC antibody was purified from hybridomas clone 9E10 and used at 1/500 (ATCC® CRL-1729, ATCC France). The p27kip1 antibody (C19) was purchased from Santa Cruz and used at 1/1,000. The cleaved caspase-3 (Asp175) antibodies was purchased from Cell Signaling (#9661) and used at 1/500. The β-actin antibody (AC-74) was purchased from Sigma-Aldrich and used at 1/10,000.

### Real-time quantitative PCR

Total RNA (1 μg) was used as a template for cDNA synthesis, using the GoScript™ reverse transcription kit (Promega, Madison, WI, USA). The GoTaq® qPCR 2X Master Mix (Promega, Madison, WI, USA) that include all components for quantitative PCR, except sample DNA, primers and water, was used to quantify the sixteen MYC-high signature markers. Primer lists for each transcript are available in Appendix Table S2. Reaction conditions were denaturation at 95°C for 2 min; 40 cycles of 15 s at 95°C, 45 s at 60°C. Reactions were carried out using the AriaMx real-time PCR system and analyzed using the AriaMx software v1.1 (Agilent Technologies, Santa Clara, CA, USA).

### Ki67 staining and quantification

Full-thickness, 5-μm sections were cut from formalin-fixed, paraffin-embedded blocks from all 55 PDX. The samples were then stained with the Ki67 antibody (MIB-1 clone, 1:160; Dako, France) using tonsillar tissue as a positive control. Negative controls were run simultaneously with the primary antibody replaced with a buffer. Antigen retrieval was conducted in citrate buffer at pH 6.0 under pressure for 3 min. Envision Dual Link Kit (Dako) was used for detection, with diaminobenzidine (DAB) as the chromogen and hematoxylin as the counterstain. Staining was considered positive when brown nuclear labeling was observed in epithelial compartment. A standard Olympus BX41 microscope (Olympus Corp., Tokyo, Japan) was used to identify tumor hot spots in each case. The percentage of tumor cell staining was independently counted by three reviewers (BB, MB, and ND) and with "eyeballing" methodology.

### Differentiation scoring

Formalin-fixed tumors were submitted to the Hôpital Nord histology core facility, and paraffin-embedded sections were cut for hematoxylin and eosin (H&E) staining. For histopathologic scoring, H&E-stained slides were scored for the penetrance of each histological hallmark on a scale of 0–2. The predominant tumor phenotype gave the pathological score for the whole tumor (0 = poorly differentiated, 1 = moderately differentiated, 2 = well differentiated).

### Method for normalizing expression ratios

Considering the expression of upregulated gene $i$ in patient $\alpha$ as $u_{i\alpha}$ and the expression of downregulated gene $j$ in patient $\alpha$ as $d_{j\alpha}$. We calculate the sum of expression of each marker in all patients:

$$U_i = \sum_\alpha u_{i\alpha}, \quad 1 < i < 10 \quad \text{and} \quad D_j = \sum_\alpha d_{j\alpha}, \quad 1 < i < 6$$

The mean centered normalized expression was then calculated for the two set of markers:

$$\widetilde{U_{i\alpha}} = \frac{U_{i\alpha} \cdot (100)}{U_i} \quad \text{and} \quad \widetilde{D_{j\alpha}} = \frac{D_{j\alpha} \cdot (100)}{D_j}$$

The normalized ratio between upregulated gene $U_i$ and downregulated gene $D_j$ is:

$$\widetilde{r_{ij\alpha}} = \left( \frac{\widetilde{U_{i\alpha}}}{\widetilde{D_{j\alpha}}} \right), \quad 1 < i < 10 \text{ and } 1 < j < 6$$

The median of the 60 normalized ratios is then calculated:

$$\widetilde{m_\alpha} = median\left(\widetilde{r_{ij\alpha}}, \forall 1 < i < 10 \text{ and } 1 < j < 6\right)$$

$\widetilde{m_\alpha} > 1$ show a MYC-high profile and $\widetilde{m_\alpha} < 1$ show a MYC-low profile.

### Treatment with JQ1 of PDX with MYC-high and MYC-low phenotype *in vivo*

We transplanted four PDX samples with the MYC-high phenotype (CRCM16, CRCM04, CRCM114, and CRCM116) and four with the MYC-low phenotype (CRCM05, CRCM10, CRCM109, and CRCM112) subcutaneously into 6-week-old male Swiss nude mice (Crl: Nu(lco)-Foxn1nu, Charles River, Wilmington, MA). Each PDX sample was inoculated into six nude NMRI mice who were randomized for treatment ($n = 3$) and vehicle ($n = 3$). When the PDX reached 100 mm³, we started the treatment with JQ1 or vehicle and followed their growth for 28 days. JQ1 was prepared according to published procedures (Filippakopoulos *et al*, 2010) and administered intraperitoneally every day at 50 mg/kg. Tumor size was measured with a Vernier caliper twice weekly, and the tumor volume was calculated with the equation $v = (\text{length}/\text{width}^2)/2$.

### Statistical analysis

Overall survival and relapse-free survival were analyzed using the Kaplan–Meier log-rank test to assess differences in survival. Box-and-whisker plots show the medians, quartiles, and range of continuous data to demonstrate the variability of data and the degree of normality. For continuous variables, non-parametric unpaired two-tailed *t*-test was performed under the assumption of equal variance.

**Expanded View** for this article is available online.

## The paper explained

### Problem

One strategy to discover potential markers for patients stratification is to focus on identifying pathways that are deregulated in tumors, particularly when tumor cells absolutely depend of keeping these alterations (e.g., oncogene "dependence" to survive and grow). Consequently, it is logical to assume that blockage of these pathways with specific inhibitors should lead to cell growth arrest, death, and tumor regression. Using this rational, it would be possible to select, by means of a few markers, a particular subgroup of patients "addicted" to a distinct pathways, a major goal of modern individualized medicine.

### Results

To define classifier expression signature, we selected a group of 10 c-MYC target transcripts whose expression is increased in the MYC-high group and six transcripts increased in the MYC-low group. We validated the ability of these markers panel to identify MYC-high patient-derived xenografts from both: discovery and validation cohorts as well as primary cell cultures from the same patients. We then showed that cells from MYC-high patients are more sensitive to BET inhibitor treatment compared to MYC-low cells, in both monolayers, 3D cultured spheroids and in xenografted tumors, due to cell cycle arrest followed by apoptosis.

### Impact

The results provide new markers and potentially novel therapeutic modalities for distinct subgroups of pancreatic tumors and may find application to the future management of these patients within the setting of individualized medicine clinics.

## Acknowledgements

We thank Emilie Nouguerede, Jihane Pakradouni and members of the cell culture platform (CRCM, Unité 1068) for their technical assistance. This work was supported by La Ligue Contre le Cancer, INCa, Canceropole PACA, DGOS (labellisation SIRIC) and INSERM to JI.

## Author contributions

BB, MB, OG, CL, AM, RN, YB, LM, and AM performed experiments. MGil, VM, OT, J-RD, MGio, PG, MGas, MO, JEB, and JQ provided material. SG, VS, MR, and FP performed the histochemical stainings and immunohistochemical scoring. J-LR, GL, RU, AS, ND, and JI performed or coordinated the bioinformatic and biostatistical analysis. RU, ND, and JI contributed to the experimental design, data analysis, and discussion. ND and JI directed the project. BB, RU, ND, and JI wrote the manuscript.

## Conflict of interest

The authors declare that they have no conflict of interest.

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
