## [Review Process File · EMBO Molecular Medicine]

Gene expression profiling of patient-derived pancreatic cancer xenografts predicts sensitivity to the BET bromodomain inhibitor JQ1: implications for individualized medicine efforts

Benjamin Bian, Martin Bigonnet, Odile Gayet, Celine Loncle, Aurélie Maignan, Marine Gilabert, Vincent Moutardier, Stephane Garcia, Olivier Turrini, Jean-Robert Delpero, Marc Giovannini, Philippe Grandva, Mohamed Gasmı, Mehdi Ouaisı, Veronique Secq, Flora Poizat, Rémy Nicolle, Yuna Blum, Laetitia Marisa, Marion Rubis, Jean-Luc Raoul, James E Bradner, Jun Qi, Gwen Lomberk, Raul Urrutia, Andres Saul, Nelson Dusetti, Juan Iovanna

Corresponding authors: Nelson Dusetti and Juan Iovanna, Centre de Recherche en Cancérologie de Marseille (CRCM), INSERM U1068, CNRS UMR 7258

Review timeline:

Submission date:	22 August 2016
Editorial Decision:	10 October 2016
Revision received:	28 November 2016
Editorial Decision:	21 December 2017
Revision received:	12 January 2017
Editorial Decision:	13 January 2017
Revision received:	25 January 2017
Accepted:	30 January 2017

Transaction Report:

Editor: Céline Carret

1st Editorial Decision

10 October 2016

Thank you for the submission of your manuscript to EMBO Molecular Medicine. We have now heard back from the three referees whom we asked to evaluate your manuscript. Although the referees find the study to be of potential interest, they also raise several issues that must be addressed in the next final version of your article.

You will see in the comments pasted below, that all three referees find the study potentially important and timely. However, the three reports are remarkably similar and raise overlapping concerns that limit the attractiveness of the paper. Of particular major interest for EMBO Mol Med scope and interests is the need for *in vivo* mice treatment with JQ1, further analysis of the genomics data, and overall provision of additional explanations and clarifications and better literature citing. Referee 3 also would like to see more mechanism, although should you be able to perform *in vivo* treatment in a satisfactory fashion, we would not insist too much on the latter.

Given the balance of these evaluations, we feel that we can consider a revision of your manuscript if

you can address the issues that have been raised within the space and constraints outlined. Please note that it is EMBO Molecular Medicine policy to allow only a single round of revision and that, as acceptance or rejection of the manuscript will depend on another round of review, your responses should be as complete as possible.

***** Reviewer's comments *****

Referee #2 (Remarks):

The authors have established a PDX collection of pancreatic tumors, which they now want to use to establish new therapeutic modalities. Here they look at the role of MYC, following multiple lines of evidence that MYC is a driver of PDAC. A relevant paper showing that MYC is haploinsufficient for RAS-driven PDAC is not quoted (Walz *et al.*, Nature). They use two published signatures of MYC target genes to stratify pancreatic tumors into two subgroups, which they term MYC-high and MYC-low group. They go on to correlate the distinction between both groups with biological parameters such as proliferation and lack of differentiation. Much of this part is circular in logic since the MYC target gene set used is highly enriched for cell cycle genes. In particular, the selected list of 10 genes is essentially a cycle gene set. Unfortunately, no further information about both groups is provided: e.g. sequences of exomes or of a panel of genes to see what mutations correlate with MYC status. Also, no molecular analysis of the molecular status of the MYC network is provided, so whether for example there is enhanced MYC binding to the genes shown in MYC high tumors is not analyzed. They then show that MYC high tumors are more sensitive to the bromodomain inhibitor JQ1, which by now is well established to inhibit MYC expression in settings where MYC is driven by a superenhancer. They show a limited analysis of JQ1 responses.

The data shown are of high quality and they represent a significant effort. While the concept has been well established, the data will spur clinical trials of bromodomain inhibitors in this entity. The enthusiasm is dampened by the limited depth of the analysis and by the fact that treatment has not been performed *in vivo* in established tumors. In my view, sequences of exomes and a better analysis of MYC function would also be required for publication.

Referee #3 (Remarks):

This paper addresses the challenge of subclassifying pancreatic adenocarcinomas in order to predict sensitivity to certain treatments. This goal is good due to clear heterogeneity between tumors and the current inability to make sense of that heterogeneity. The authors take the approach that Myc status may be useful, owing to its central role in tumor biology. Using genes that previously were determined as part of the Myc pathway, the authors derived a myc-high and myc-low grouping, and then tested for differences between the groups in various characteristics including response to the BET inhibitor JQ1. The paper presents interesting data suggestive of some value in the classification. This work has good potential, but I believe some important points should be addressed, as described below.

In the development of a predictive signature, normally the training set is defined by a "gold standard," such as diagnosis, outcome, or survival. Here the training set was simply the division of the tumors after hierarchical clustering. When one clusters samples, the samples always will fall into two groups by necessity; the grouping may or may not have any significance. The authors do show that differences exist between the groups in survival and response to BET inhibitors, which is interesting, but the testing of the predictive capability of the signature does not go far enough. The ability of the signature to predict sensitivity to BET inhibitors was tested on only 8 cultures in the

training set and 6 cultures in the test set. These numbers are too small to determine the predictive ability of the signature.

If the final goal is to identify predictors of response to BET inhibitors, it seems a better approach would be to divide the training set of pdx models by response to BET inhibitor, develop a classifier based on that grouping, and then test the predictive ability of the classifier on new pdx models. Perhaps such an approach would give more accurate predictions than the method presented in the manuscript.

Why did the authors not treat the mice with the BET inhibitors, rather than using 2D cultures? Then all the models could be tested.

The authors developed a 16-gene classifier for the grouping defined by the hierarchical clustering of 239 genes. The method of classification was based on ratios between genes. Previous methods of classifying samples used logistic regression, recursive partitioning, or related methods. The authors could explain why they chose the ratio method and whether they compared it to other classification methods.

The authors evaluated the growth rate of the tumors in the training set using semi-quantitative ki67 scoring. Did they evaluate the growth rates of the tumors in the mice? Do the growth rates show differences between the groups?

In the validation cohort of 16 pdx, the authors should test whether differences exist between the 2 groups in outcome, growth rate and differentiation, as they did for the training set. The histological determination of the differentiation should be done blinded to the myc status; the authors should comment on whether that method was used in both the training and test sets.

Other comments:

First paragraph of Results section: Which anatomopathological characteristics were preserved, and how was that determined? In how many successive passages were these characteristics determined?

Fig. 4B, 4C, and 4D: With only 4 samples per group, individual points could be shown instead of columns, as was done in fig. 5C.

In the introduction, the authors should describe whether previous attempts have been made to predict response to JQ1, and also why a more direct measure of Myc activity is not sufficient. Why is an indirect gene expression signature needed?

Referee #4 (Remarks):

Bian et al. developed a novel RNA expression-based identifier to characterize pancreatic cancer with high MYC expression. Furthermore, the author's postulate that such "MYC-high" pancreatic cancers are characterized by an increased sensitivity towards the BET inhibitor JQ1. Pancreatic cancer remains a devastating disease. Development of new treatments options is an urgent medical need. Therefore, the work addresses an important aspect of current translational medical research. However, there are several limitation of current study.

Major points:

1. The author's follow a concept of synthetic lethality between specific cellular vulnerabilities and high MYC expression. E.g. it is clear that MYC induces vulnerabilities in the G2/M phase of the cell cycle and such vulnerability can be exploited with specific therapies (e.g. Topham et al. 2014; Perera & Venkitaraman, 2016). Whereas the molecular mechanisms for the G2/M phase vulnerability are at least in part characterized at the molecular level, it is completely unclear whether the BET inhibitor is triggering a certain and specific vulnerability of pancreatic cancer cells, whether the regulation of MYC upon BET inhibitor treatment is specific for JQ1 sensitive pancreatic cancers, or whether only "MYC high" pancreatic cancers are addicted to the oncogene. Deciphering the molecular underpinnings of the increased sensitivity of "MYC-high" pancreatic cancers to JQ1 is necessary and would be a clear conceptual advance.
2. The author's show that the "MYC-high" phenotype is connected to an increased proliferation

index. How can the author's exclude that the increased JQ1 sensitivity is simply connected to an increased proliferation rate? Is the proliferation rate documented in the PdX at the level of IHC (KI67) also evident in the cell lines? Are cytotoxic and targeted therapeutics generally more potent in the "MYC-high" phenotype? A specificity for JQ1 would support the author's conclusions.

3. In Fig. 1A the author's grouped PDAC into MYC high and low according to the differential expression of MYC target genes. To provide a further layer of evidence that the method sufficiently determines the MYC status, the author's should investigate the MYC expression score (nuclear staining intensity x number of positive tumor cells) in both groups at the level of IHC and compare it. Furthermore, in the cell lines models of pancreatic cancer, the author's should determine the MYC expression levels by western blot and analyze the connection to the identifier results and JQ1 sensitivity.

- For Fig. 1B, a complete list of statistically significant enriched signatures in the MYC high and low groups should be provided as a supplemental file (the complete MYC signature plot including Hallmark, PID, and curated gene set signatures). In addition, it is important to demonstrate that „classical" MYC signatures, including V1, V2, or Dang - MYC targets up, are indeed enriched in the MYC high phenotype. Furthermore, to assure robustness of their findings, the author's should demonstrate that the same signatures are enriched in different pancreatic cancer cohorts, clustered by the new identifier in the MYC-high versus MYC-low phenotype. Here, including the RNA-seq data of the TCGA and the microarray data published by Collisson et al., 2011 is important and would increase the value of the findings. Are the same signatures also enriched in these cohorts?

4. The author's suggest a stratification by MYC implicating a clinical decision for BET inhibitors in "MYC-high" pancreatic cancers. In the spheroid model, JQ1 (2 uM) reduced viability to only 50% in sensitive tumors and in the cell-based model, IC50 values between 2-6 uM were determined. Is this effect clinically relevant? JQ1 demonstrates IC50 values of only 100 nM in sensitive tumor entities. Please reconsider some statements, tune down and point to other possibilities using JQ1 (e.g. combinations).

5. Are the four false positive assigned "MYC-high" PDACs JQ1 sensitive or resistant? Is MYC protein expression high or low?

Further points:

- The author's should at least discuss, why rather classical MYC target genes (e.g. ODC1, NCL, HSPE1) are not included in the specific MYC signatures, defined by the author's.

- "A frequently deregulated, although insufficiently therapeutically exploited pathway in pancreatic cancer involves the "addiction" to c-Myc oncogene (Mertz et al, 2011)."

This topic should be discussed more carefully. Formally, the addiction of pancreatic cancer to MYC is not demonstrated so far. Furthermore, to my knowledge, the paper of Mertz et al. investigates mainly action of BET inhibitors in lymphoma and leukemia.

- "c-Myc is found to be amplified in more than 30% as well as overexpressed in more than 40% of tumors, with additional cases displaying rearrangement or changes in methylation of this locus (Dang et al, 2009)."

Please cite the current genetic data from the TCGA atlas (14% amplifications) and the sequencing study from the Knudsen lab (Witkiewicz AK et al., 2015) (approximately 12% amplifications). Especially the second study is important, since the only CNV with a clinical impact were MYC CNVs. This can be highlighted to underscore the relevance of MYC in a KRAS-driven cancer. The manuscript of Dang et al. is a nice review article in CCR. Please correct.

- "In addition, using a variety of experimental models, it has been later shown that MYC activation induces tumor growth, DNA replication, protein synthesis and increases tumor cell metabolism, angiogenesis and suppression of the host immune response (Gabay et al, 2014; Huang & Weiss, 2013; Roussel & Robinson, 2013; Schmitz et al, 2014). Moreover, MYC activates stemness, blocks cellular senescence (Bachireddy et al, 2012; Dang et al, 2009; Gamberi et al, 1998), and its overexpression is frequently associated with poor clinical outcome and aggressiveness (Nesbit et al, 1999)."

It is unclear why so many reviews or papers from other tumor entities are cited, but important work demonstrating the function of MYC in the pancreatic context are neglected. Some prominent examples include the work of the Wagner lab (Lin et al. 2013; demonstration that MYC on its own can induce PDAC), the Sansom/Eilers groups (Walz et al. 2014, deletion of one MYC allele decelerates tumor development in vivo), the Lowe group (Saborowski M et al., 2014, MYC siRNA blocks tumor development in vivo), or the Heeschen group (Sancho P et al., implication of potential problems of MYC inhibition in pancreatic cancer). The concept of MYC as a stratification marker was recently summarized (e.g. Wirth & Schneider, 2016). Also data concerning the action of JQ1 in pancreatic cancer are incompletely cited and discussed (e.g. Garcia et al., 2016; Sahai et al. 2014

papers are missing). Especially the paper of Garcia et al. should be discussed, since instead of MYC, regulation of the cell cycle regulator CDC25B by BET inhibition was connected to the sensitivity in pancreatic cancer PDX models. In line, Lowe's group found no connection of MYC expression levels to the sensitivity of HCC's to JQ1, but detected that MYC protein expression is a marker for CDK9 inhibitor sensitivity. Please, discuss (Huang et al., 2014).

- "Seventeen samples of the cohort were previously published (Duconseil et al, 2015) as GEO accession number GSE55513." Please provide an accession number for all 55 arrays.

- For figure 3 it is necessary to depict that only n=4 for each phenotype was analyzed.

1st Revision - authors' response

28 November 2016

Referee #2 (Remarks)

The authors have established a PDX collection of pancreatic tumors, which they now want to use to establish new therapeutic modalities. Here they look at the role of MYC, following multiple lines of evidence that MYC is a driver of PDAC. A relevant paper showing that MYC is haploinsufficient for RAS-driven PDAC is not quoted (Walz *et. al.*, Nature).

The study from *Walz et al. (2014)* has been cited in the second paragraph of the Introduction section.

They use two published signatures of MYC target genes to stratify pancreatic tumors into two subgroups, which they term MYC-high and MYC-low group. They go on to correlate the distinction between both groups with biological parameters such as proliferation and lack of differentiation. Much of this part is circular in logic since the MYC target gene set used is highly enriched for cell cycle genes. In particular, the selected list of 10 genes is essentially a cycle gene set. Unfortunately, no further information about both groups is provided: e.g. sequences of exomes or of a panel of genes to see what mutations correlate with MYC status.

This is an important remark made by the reviewer. In fact, this question concerns whether or not the genetic alterations of the *Myc* gene in PDAC tumors are predictive of the JQ1 response. To address this question, we investigated this clinically relevant aspect of the tumors examined in our study. Consequently, we now provide the *Myc* copy number analysis for Myc-high and Myc-low samples in Supplementary Figure S2 of the new version of the manuscript. This analysis reveals that 15 of 17 PDX samples with a Myc-high phenotype shown a gain in the *Myc* gene copy number, but surprisingly, 20 of 38 PDX from the Myc-low group also shown an increase in the CNV of *Myc* gene. Thus, in the Myc-low samples, those not responders to JQ1 compound, around a half of patients shown also an increase in *Myc* copy number. This is likely the result of epigenomics compensations that are triggered upon Myc amplifications, similar to those that have been reported in transgenic mice which have multiple insertions of the exogenous construct but express low levels of the gene in question. Based on this observation we can conclude that the *Myc* copy number alteration is an unsuitable prediction method to estimate the BET inhibitors sensitivity.

Also, no molecular analysis of the molecular status of the MYC network is provided, so whether for example there is enhanced MYC binding to the genes shown in MYC high tumors is not analyzed.

The reviewer makes a very interesting suggestion for understanding the molecular function of the MYC in PDAC. However, the main goal of our study is to define a realistic signature as well as possible and overall that it became technically applicably for the patients. In our mind, expression of a limited number of transcripts reflecting the MYC activity of the PDAC should be measured in small samples of PDAC obtained by EUS-FNA, an approach that is used in almost all patients before starting the treatment. Our preliminary data on samples obtained by EUS-FNA confirm this conjecture.

They then show that MYC high tumors are more sensitive to the bromodomain Inhibitor JQ1, which by now is well established to inhibit MYC expression in settings where MYC is driven by a superenhancer. They show a limited analysis of JQ1 responses.

The data shown are of high quality and they represent a significant effort. While the concept has been well established, the data will spur clinical trials of bromodomain inhibitors in this entity. The

enthusiasm is dampened by the limited depth of the analysis and by the fact that treatment has not been performed *in vivo* in established tumors.

This is shared concern from reviewers #2 and #3. We agree completely with this point. Consequently, in a collaborative effort with the Bradner's lab, we performed these *in vivo* experiments on 4 Myc-high and 4 Myc-low samples. Data was presented in Figure 6 of the new version of the manuscript. As expected, and according to the results obtained on cell lines and spheroids, Myc-high samples are significantly more sensitive to JQ1 treatment.

In my view, sequences of exomes and a better analysis of MYC function would also be required for publication.

As mentioned above, we performed exome sequencing analyses on 29 of the 55 samples of this study and although a few number of *Myc* mutations were identified they are not indicative of the Myc activity of the PDAC. In addition, not association of *Myc* status with other mutations (*Kras*, *SMAD4*, *p53* or *INK4*) and JQ1 sensitivity was found in these samples. We conclude that Myc transcriptional activity is under a complex system of regulation, probably more than previously assumed.

Referee #3 (Remarks)

This paper addresses the challenge of subclassifying pancreatic adenocarcinomas in order to predict sensitivity to certain treatments. This goal is good due to clear heterogeneity between tumors and the current inability to make sense of that heterogeneity. The authors take the approach that Myc status may be useful, owing to its central role in tumor biology. Using genes that previously were determined as part of the Myc pathway; the authors derived a myc-high and myc-low grouping, and then tested for differences between the groups in various characteristics including response to the BET inhibitor JQ1. The paper presents interesting data suggestive of some value in the classification. This work has good potential, but I believe some important points should be addressed, as described below.

In the development of a predictive signature, normally the training set is defined by a "gold standard," such as diagnosis, outcome, or survival. Here the training set was simply the division of the tumors after hierarchical clustering. When one clusters samples, the samples always will fall into two groups by necessity; the grouping may or may not have any significance.

We thank to this reviewer for her/his logical comment and have taken into consideration this rational. To validate our strategy, in this paper, we used two complementary approaches: the first one analyzed the differences between both myc-high and myc-low groups with a stringent statistical analysis and found that the p values for the 10 top transcripts overexpressed in myc-high is = 0.00196 with a FDR of 0.004458 (Supplementary Table S1). The second approach validated our myc-associated signature using an independent "test" group in a prospective manner. Thus, we believe that applying these complementary strategies the results should be very confident.

The authors do show that differences exist between the groups in survival and response to BET inhibitors, which is interesting, but the testing of the predictive capability of the signature does not go far enough. The ability of the signature to predict sensitivity to BET inhibitors was tested on only 8 cultures in the training set and 6 cultures in the test set. These numbers are too small to determine the predictive ability of the signature.

This remark by the reviewer is also logic but, in our opinion, the exact number of samples to estimate the prediction quality of a signature is very difficult to establish *a priori*. We applied a statistical analysis for comparing the response of both myc-high and myc-low groups in the training group and found a significant difference among them in terms of cell viability (Fig 4B) or IC50 (Fig 4C) as well as in the test group (Fig 5C and 5D). We based our confidence on those analyses.

If the final goal is to identify predictors of response to BET inhibitors, it seems a better approach would be to divide the training set of pdx models by response to BET inhibitor, develop a classifier based on that grouping, and then test the predictive ability of the classifier on new pdx models.

Perhaps such an approach would give more accurate predictions than the method presented in the manuscript.

We agree with this reviewer if the goal of our work was exclusively to identify patients sensitive to BET inhibitors. But our goal was not exactly that but rather to select patients with high myc activity in their PDAC, which could be better responders to BET inhibitors. This is why we preferred to select these samples through a myc signature rather than a hypothetical BET inhibitors-associated signature. This approach adds potential “clinical utility” to our study.

Why did the authors not treat the mice with the BET inhibitors, rather than using 2D cultures? Then all the models could be tested.

As mentioned above, this remark is shared between reviewers #2 and #3. We agree with their point. Thus, in a collaborative effort with the Bradner’s lab, we performed these *in vivo* experiments on 4 Myc-high and 4 Myc-low samples. Data was presented in Figure 6 of the new version of the manuscript. As expected, and according to the results obtained on cell lines and spheroids, Myc-high samples are significantly more sensitive to JQ1 treatment.

The authors developed a 16-gene classifier for the grouping defined by the hierarchical clustering of 239 genes. The method of classification was based on ratios between genes. Previous methods of classifying samples used logistic regression, recursive partitioning, or related methods. The authors could explain why they chose the ratio method and whether they compared it to other classification methods.

This is also a pertinent observation provided by the reviewer. An important challenge for phenotype classification using gene expression data is to develop techniques that not only yield accurate and robust decision rules, but they are also easy to interpret. Advanced statistical learning and pattern recognition methods are routinely applied to transcriptomics and other high-throughput data. These include neural networks, decision trees, boosting, and support vector machines. In many cases, these methods achieve good classification performance, with sensitivities and specificities above ninety percent. However, they generally result in extremely complex decision rules based on nonlinear functions of many gene expression values. Therefore, whereas advanced methods may be more accurate than those based on the patterns of individual genes, they usually produce decision rules which are virtually impossible to interpret. Furthermore, as the number of variables (transcripts) far exceeds the number of observations in most microarray studies, building more complex classifiers entails a greater risk of over-fitting the training data and poor generalization. Here we focus on relative expression analysis methods which involve a small number of gene pairs, each exhibiting a characteristic “relative expression reversal” between the phenotypes or classes of interest. Aggregating the decisions from a few such pairs is surprisingly powerful. In this study, we used gene-pair relative expression markers and specifically in the form of a two-gene expression-level ratio. This kind of approach has been previously used for disease classification and prognosis. For example Gordon et al. successfully distinguished between malignant pleural mesothelioma and adenocarcinoma of the lung based on ratios of expression (Gordon GJ, et al. *Cancer Res.* 2002: Translation of microarray data into clinically relevant cancer diagnostic tests using gene expression ratios in lung cancer and mesothelioma. *Cancer Res.* 2002;62:4963-4967).

The authors evaluated the growth rate of the tumors in the training set using semi-quantitative ki67 scoring. Did they evaluate the growth rates of the tumors in the mice? Do the growth rates show differences between the groups?

The present work clearly demonstrates that myc-high and myc-low samples have some different behavior, primarily concerning their growth rates. As requested by this reviewer, we evaluated the growth rate of xenografts in mice and in the doubling time of derived primary cell lines. As expected, the myc-high xenografts growth more rapidly than the myc-low (see below). However, their cell doubling time, as measured *in vitro*, on their derived cells results similar (see below). However, as showed in this paper, Myc-high cells *in vitro* and Myc-high PDX *in vivo* are more sensitive to the BET inhibitor treatment. These data suggest that JQ1 efficiency of JQ1 is dependent of the Myc activity instead cells growth rate.

In the validation cohort of 16 pdx, the authors should test whether differences exist between the 2 groups in outcome, growth rate and differentiation, as they did for the training set. The histological determination of the differentiation should be done blinded to the myc status; the authors should comment on whether that method was used in both the training and test sets.

Respectfully, contrary to the “training” group that needs to be strongly characterized, we assume that the “testing” group is only necessary for validating the obtained signature. Moreover, the testing group, as presented in this work, is mimicking the clinic in which a patient should be selected as Myc-high or Myc-low without any other parameter than its own signature. In addition, giving a table with clinical outcome, growth rate and differentiation of the testing group will result redundant with the testing group.

Other comments:

First paragraph of Results section: Which anatomopathological characteristics were preserved, and how was that determined?

The anatomopathological characteristics mentioned were: 1/ Nucleocytoplasmic ratios, nuclei alignment, eosinophilia, differentiation degree analysis by H&E staining; 2/ Presence of mucins revealed by alcian blue staining. Those characteristics were mentioned in the main text of the new version of the manuscript.

In how many successive passages were these characteristics determined?

The main anatomopathological characteristics of patient primary tumors were preserved in xenografts for at least 6 successive passages as previously described in Duconseil et al., 2015. A sentence describing this fact was included in the result Section of the new version of the manuscript.

Fig. 4B, 4C, and 4D: With only 4 samples per group, individual points could be shown instead of columns, as was done in fig. 5C.

We value the opinion that when possible, the representation of data as boxes and whiskers is clearer and more comprehensive for readers. However, for representing data as boxes and whiskers the numbers of samples must be 4 or more (as is the case in Figure 4) but we have only 3 values in the case of Figure 5 and therefore we are forced to represent data as individual points and its median value. We hope the reviewer understand our position and share our point of view.

In the introduction, the authors should describe whether previous attempts have been made to predict response to JQ1, and also why a more direct measure of Myc activity is not sufficient. Why is an indirect gene expression signature needed?

We are grateful to this reviewer for underlining this point. A pertinent paragraph has been added to the introduction to explain why it is necessary to identify a transcriptional signature associated to myc activity to select PDAC sensitive to BET inhibitors. The paragraph is the following: “Several studies have focus on the discovery of predictive markers of response to BET inhibitors. Puissant et al. reported that amplification of MYCN in medulloblastoma was the most robust marker for

predicting the sensitivity of those tumors to JQ1 (Puissant et al, 2013). Moreover, certain rare tumors called NUT midline carcinomas carrying tandem fusion of BRD4 and NUT genes (nuclear protein in testis), show an important sensitivity to BET inhibitors (Stathis et al, 2016). However, in addition to these relatively rare examples it was very difficult to predict a response to the BET inhibitors by genomic approaches. To overcome this issue, the use of tumoral transcriptional program can be an effective way to develop and characterize robust predictive signatures notably in term of chemosensitivity". We hope that this explanation satisfies the reviewer.

Referee #4 (Remarks):

Bian et al. developed a novel RNA expression-based identifier to characterize pancreatic cancer with high MYC expression. Furthermore, the author's postulate that such "MYC-high" pancreatic cancers are characterized by an increased sensitivity towards the BET inhibitor JQ1. Pancreatic cancer remains a devastating disease. Development of new treatments options is an urgent medical need. Therefore, the work addresses an important aspect of current translational medical research. However, there are several limitation of current study.

Major points:

1. The author's follow a concept of synthetic lethality between specific cellular vulnerabilities and high MYC expression. E.g. it is clear that MYC induces vulnerabilities in the G2/M phase of the cell cycle and such vulnerability can be exploited with specific therapies (e.g. Topham et al. 2014; Perera & Venkitaraman, 2016). Whereas the molecular mechanisms for the G2/M phase vulnerability are at least in part characterized at the molecular level, it is completely unclear whether the BET inhibitor is triggering a certain and specific vulnerability of pancreatic cancer cells, whether the regulation of MYC upon BET inhibitor treatment is specific for JQ1 sensitive pancreatic cancers, or whether only "MYC high" pancreatic cancers are addicted to the oncogene. Deciphering the molecular underpinnings of the increased sensitivity of "MYC-high" pancreatic cancers to JQ1 is necessary and would be a clear conceptual advance.

This remark is important, interesting and pertinent but relatively out of the scope of this work. In fact, apparently, a subgroup of PDAC samples with high MYC activity seems to be more sensitive to the treatment with BET inhibitors and, consequently, they should be therapeutically targeted with this type of compounds, as we hypothesized. Conversely, however, myc-low samples seem to be less sensitive, but not completely insensitive, as showed in Figures 4, 5 and 6, which have a sense. This last group should be also treated with BET inhibitors but we presume that their sensitivity will be significantly lesser and therefore they will be non- or bad-responders.

2. The author's show that the "MYC-high" phenotype is connected to an increased proliferation index. How can the author's exclude that the increased JQ1 sensitivity is simply connected to an increased proliferation rate? Is the proliferation rate documented in the PdX at the level of IHC (KI67) also evident in the cell lines? Are cytotoxic and targeted therapeutics generally more potent in the "MYC-high" phenotype? A specificity for JQ1 would support the author's conclusions.

This is an obvious and pertinent comment. As requested by reviewer 2 (see above), we measured the cell doubling time of some myc-high and myc-low derived cell lines and found that *in vitro* they don't show significant differences in their growth curves whereas, on the contrary, they present significantly different in their sensitivity to the JQ1 compound. From these observations we assume that the growth ratio is not a major factor for the JQ1 sensitivity.

3. In Fig. 1A the author's grouped PDAC into MYC high and low according to the differential expression of MYC target genes. To provide a further layer of evidence that the method sufficiently determines the MYC status, the author's should investigate the MYC expression score (nuclear staining intensity x number of positive tumor cells) in both groups at the level of IHC and compare it. Furthermore, in the cell lines models of pancreatic cancer, the author's should determine the MYC expression levels by western blot and analyze the connection to the identifier results and JQ1 sensitivity.

This is also an interesting point raised by this reviewer. In this regard, although the level of MYC in PDAC samples could be associated to an increased MYC activity, this is not systematic since its transcriptional activity is dependent, on one hand, of its post-translational modifications and, on the

other hand, of can be compensated by unknown epigenomic mechanisms. This is why the rational to study the expression of the myc targeted genes seems to be the more consistent alternative to determine its activity.

- For Fig. 1B, a complete list of statistically significant enriched signatures in the MYC high and low groups should be provided as a supplemental file (the complete MYC signature plot including Hallmark, PID, and curated gene set signatures). In addition, it is important to demonstrate that „classical" MYC signatures, including V1, V2, or Dang - MYC targets up, are indeed enriched in the MYC high phenotype. Furthermore, to assure robustness of their findings, the author's should demonstrate that the same signatures are enriched in different pancreatic cancer cohorts, clustered by the new identifier in the MYC-high versus MYC-low phenotype. Here, including the RNA-seq data of the TCGA and the microarray data published by Collisson et al., 2011 is important and would increase the value of the findings. Are the same signatures also enriched in these cohorts?

This is an interesting suggestion. A complete list of statistically significant enriched signatures using Biological Process, Curated Geneset Enriched and Hallmarks Enriched tools was performed and presented in Supplementary Tables 3a, 3b, 3c, 3d, 3e and 3f of the new version of the manuscript. The second point is a little difficult to be interpreted, because as we used V1 and V2 list of genes to select the groups of patients, these genes will be automatically enriched in the Myc-high group.

4. The author's suggest a stratification by MYC implicating a clinical decision for BET inhibitors in "MYC-high" pancreatic cancers. In the spheroid model, JQ1 (2 uM) reduced viability to only 50% in sensitive tumors and in the cell-based model, IC50 values between 2-6 uM were determined. Is this effect clinically relevant? JQ1 demonstrates IC50 values of only 100 nM in sensitive tumor enties. Please reconsider some statements, tune down and point to other possibilities using JQ1 (e.g. combinations).

The point raised by the reviewer is correct. Spheroids volume as measured in this work does not take into account the cell viability but only their volume as indicated in the Figure 4. In this experimental context dead cells remain integrated in spheroids and therefore are considered to be part of their volume. This is not the case when viability is measured with appropriate reagents. In this case, dead cells were not considered. This is why differences between effect of treated and non-treated spheroids is underestimated in Figure 4. Another point underlined by this reviewer is concerning the clinical relevance of JQ1. We performed *in vivo* experiments as asked by reviewers 2 and 3 and found that, as expected, myc-high PDX are more sensitive than myc-low PDX which is strongly supporting our hypothesis. In any case we are suggesting to treating patients with BET inhibitors exclusively; in fact we only propose is to selecting patients with a myc-high phenotype for to be treated with BET inhibitors with the idea to optimize its antitumor effect. The text of the manuscript was modified accordingly.

5. Are the four false positive assigned "MYC-high" PDACs JQ1 sensitive or resistant? Is MYC protein expression high or low?

This is an important observation brought about by this reviewer. We were unable to produce primary cells from the CRCM100 PDX and, thus unfortunately, cannot address his/her concern for this sample. However, for CRCM06, CRCM12 and CRCM27 the IC50 were 5 µM, 23 µM and 14 µM, respectively. In conclusion, CRCM12 and CRCM27 are relatively resistant but the CRCM06 is sensible.

Further points:

- The author's should at least discuss, why rather classical MYC target genes (e.g. ODC1, NCL, HSPE1) are not included in the specific MYC signatures, defined by the author's.

As stated under Material and Methods section we only selected myc targets genes included in both V1 and V2 sets of Myc-dependent genes. In these sets of genes, ODC1 was present, overexpressed in myc-high subgroup as expected, but it is not at the top of differentially expressed genes. NCL and HSPE1 were not included in these reference sets of genes.

- "A frequently deregulated, although insufficiently therapeutically exploited pathway in pancreatic cancer involves the "addiction" to c-Myc oncogene (Mertz et al, 2011)."

This topic should be discussed more carefully. Formally, the addiction of pancreatic cancer to MYC is not demonstrated so far. Furthermore, to my knowledge, the paper of Mertz et al. investigates mainly action of BET inhibitors in lymphoma and leukemia.

The sentence was modified according to the reviewer.

- "c-Myc is found to be amplified in more than 30% as well as overexpressed in more than 40% of tumors, with additional cases displaying rearrangement or changes in methylation of this locus (Dang et al, 2009)."

Please cite the current genetic data from the TCGA atlas (14% amplifications) and the sequencing study from the Knudsen lab (Witkiewicz AK et al., 2015) (approximately 12% amplifications). Especially the second study is important, since the only CNV with a clinical impact were MYC CNVs. This can be highlighted to underscore the relevance of MYC in a KRAS-driven cancer. The manuscript of Dang et al. is a nice review article in CCR. Please correct.

References were changed accordingly to the reviewer.

- "In addition, using a variety of experimental models, it has been later shown that MYC activation induces tumor growth, DNA replication, protein synthesis and increases tumor cell metabolism, angiogenesis and suppression of the host immune response (Gabay et al, 2014; Huang & Weiss, 2013; Roussel & Robinson, 2013; Schmitz et al, 2014). Moreover, MYC activates stemness, blocks cellular senescence (Bachireddy et al, 2012; Dang et al, 2009; Gamberi et al, 1998), and its overexpression is frequently associated with poor clinical outcome and aggressiveness (Nesbit et al, 1999)."

It is unclear why so many reviews or papers from other tumor entities are cited, but important work demonstrating the function of MYC in the pancreatic context are neglected. Some prominent examples include the work of the Wagner lab (Lin et al. 2013; demonstration that MYC on its own can induce PDAC), the Sansom/Eilers groups (Walz et al. 2014, deletion of one MYC allele decelerates tumor development in vivo), the Lowe group (Saborowski M et al., 2014, MYC siRNA blocks tumor development in vivo), or the Heeschen group (Sancho P et al., implication of potential problems of MYC inhibition in pancreatic cancer). The concept of MYC as a stratification marker was recently summarized (e.g. Wirth & Schneider, 2016). Also data concerning the action of JQ1 in pancreatic cancer are incompletely cited and discussed (e.g. Garcia et al., 2016; Sahai et al. 2014 papers are missing). Especially the paper of Garcia et al. should be discussed, since instead of MYC, regulation of the cell cycle regulator CDC25B by BET inhibition was connected to the sensitivity in pancreatic cancer PdX models. In line, Lowe's group found no connection of MYC expression levels to the sensitivity of HCC's to JQ1, but detected that MYC protein expression is a marker for CDK9 inhibitor sensitivity. Please, discuss (Huang et al., 2014).

References were modified accordingly to suggestions of the reviewer.

- "Seventeen samples of the cohort were previously published (Duconseil et al, 2015) as GEO accession number GSE55513." Please provide an accession number for all 55 arrays.

The 38 affymetrix datasets (in duplicate) were deposited in the GEO database with the accession number GSE89792

- For figure 3 it is necessary to depict that only n=4 for each phenotype was analyzed.

The number of samples was included in Figure 3.

Thank you for the submission of your revised manuscript to EMBO Molecular Medicine. We have now received the enclosed reports from the referees that were asked to re-assess it. As you will see the reviewers are now globally supportive but before we consider moving forward, please address the remaining concerns as thoroughly as possible. Please also add the following final amendments:

1) Please address the comments provided by referees 3 and 4. Please provide a letter INCLUDING the reviewer's reports and your detailed responses to their comments (as Word file).

2) In the main manuscript file:

- M&M: please confirm that the SNP array data were deposited under the GEO accession numbers GSE55513 and GSE89792; if not please provide a novel accession number.

Please submit your revised manuscript within one month. I look forward to seeing a revised form of your manuscript as soon as possible.

***** Reviewer's comments *****

Referee #3 (Comments on Novelty/Model System):

the idea of finding of Myc signature and testing its relevance is novel I think and potentially valuable.

Referee #3 (Remarks):

In general I am satisfied with the explanations given by the authors and the revisions. The addition of the mouse data showing treatment with JQ1 is helpful.

I am not satisfied with (or I don't fully understand) the authors' response to the point that begins "In the validation cohort of 16 pdx..." My point was that if the authors are claiming that differences exist in the training set in outcome, growth rate, and differentiation, then naturally the question arises as to whether the differences are real and repeatable, or just random phenomena observed only in the training set. So the point was just to determine whether the differences previously observed are real, not necessarily to characterize the samples. But I will leave it to the authors to follow up on this point.

The other additions to text are good and I believe strengthen the manuscript.

An additional note: In their responses to comments, it would be helpful to more fully describe where changes or explanations appear in the text.

Referee #4 (Remarks):

The manuscript has improved and documents a possibility to stratify for more BET inhibitor sensitive pancreatic cancers. However, in my view, it remains important to connect the "MYC classifier" to MYC protein expression to increase robustness. I agree with the authors that numerous mechanisms regulate MYC activity, however the protein should be expressed to a certain extent and the protein executes activity. This note is especially important in pancreatic cancer, where a broad range of MYC expression levels, ranging from below the detection level in IHC to very high expression, is documented. Therefore, the demonstration that MYC classifier high and JQ1 sensitive PDXs show a high nuclear MYC expression score is important. Furthermore, there are several ways to enrich for active MYC in cell based models (e.g. analyzing the chromatin bound fraction in western blots). Such information is important even if MYC protein expression is not connected to the classifier, since it influences potential stratification concepts and technologies.

Further points:

- "This phenomenon suggests that potential epigenetic mechanisms are deployed by cells to compensate for the increase in MYC copy number in this tumor." Beyond epigenetics, numerous

other mechanisms might contribute here. Please tune down.

- The author's state that MYC high PDX „efficiently responded". Please describe the response more precisely and the term "temporary tumor stasis" might by an alternative to describe the response. How were the 8 PDX in vivo models (n=4 classifier high, n=4 classifier low) selected out of the groups of 17 classifier high and 38 classifier low PDX?
- Please define "CSCs"
- Please correct "transcriptionnal"
- Please adapt the y axis of S2

2nd Revision - authors' response

12 January 2017

[Response to Editor]

- M&M: please confirm that the SNP array data were deposited under the GEO accession numbers GSE55513 and GSE89792; if not please provide a novel accession number.

- *GSE55513 is the first microarray dataset of 17 patients; GSE89792 is the second dataset for the next set of patients. The SNP array dataset was deposited under the accession number: E-MTAB-5006 (<http://www.ebi.ac.uk/arrayexpress>)*

[Response to Reviewers]

Referee #3 (Comments on Novelty/Model System):

The idea of finding of Myc signature and testing its relevance is novel I think and potentially valuable.

Referee #3 (Remarks):

In general I am satisfied with the explanations given by the authors and the revisions. The addition of the mouse data showing treatment with JQ1 is helpful.

I am not satisfied with (or I don't fully understand) the authors' response to the point that begins "In the validation cohort of 16 pdx..." My point was that if the authors are claiming that differences exist in the training set in outcome, growth rate, and differentiation, then naturally the question arises as to whether the differences are real and repeatable, or just random phenomena observed only in the training set. So the point was just to determine whether the differences previously observed are real, not necessarily to characterize the samples. But I will leave it to the authors to follow up on this point.

First, in fact we made a mistake in the previous answer to this question and we are very sorry about that. The last sentence of our answer was:

*« In addition, giving a table with clinical outcome, growth rate and differentiation of the testing group will result redundant with the **testing** group. » instead « In addition, giving a table with clinical outcome, growth rate and differentiation of the testing group will result redundant with the **learning** group. »*

In addition, the clinical data of the validation cohort concerning outcome weren't available during the revision process. At this time we have 8/16 patients (4 Myc-high and 4 Myc-low PDAC). As the reviewer may observe in the Figure we are presenting below, the Kaplan Meier curve of these patients shows a significant reduced OS in Myc-high compared to Myc-low patients.

Valiation Cohort (Overall Survival)
The other additions to text are good and I believe strengthen the manuscript.

An additional note: In their responses to comments, it would be helpful to more fully describe where changes or explanations appear in the text.

Referee #4 (Remarks):

The manuscript has improved and documents a possibility to stratify for more BET inhibitor sensitive pancreatic cancers. However, in my view, it remains important to connect the "MYC classifier" to MYC protein expression to increase robustness. I agree with the authors that numerous mechanisms regulate MYC activity, however the protein should be expressed to certain extent and the protein executes activity. This note is especially important in pancreatic cancer, where a broad range of MYC expression levels, ranging from below the detection level in IHC to very high expression, is documented. Therefore, the demonstration that MYC classifier high and JQ1 sensitive PDXs show a high nuclear MYC expression score is important. Furthermore, there are several ways to enrich for active MYC in cell based models (e.g. analyzing the chromatin bound fraction in western blots). Such information is important even if MYC protein expression is not connected to the classifier, since it influences potential stratification concepts and technologies.

We understand the point of view of this reviewer concerning that the level of MYC proteins should be connected to the PDX's MYC status (high or low). In that way, we screened the levels of MYC protein in total extract from 6 MYC-high and 6 MYC-low PDX by Western Blot.

Results reveal that no difference in total MYC protein was observed on those patients.

Further points:

- "This phenomenon suggests that potential epigenetic mechanisms are deployed by cells to compensate for the increase in MYC copy number in this tumor." Beyond epigenetics, numerous other mechanisms might contribute here. Please tune down.

This quote has been tune down in the main text. *"This phenomenon suggest that potential epigenetic mechanisms might be deployed by cells to compensate for the increase in MYC copy number in this tumor"*

- The author's state that MYC high PDX „efficiently responded". Please describe the response more precisely and the term "temporary tumor stasis" might by an alternative to describe the response.

How were the 8 PDX in vivo models (n=4 classifier high, n=4 classifier low) selected out of the groups of 17 classifier high and 38 classifier low PDX?

In this experiment we selected (according to the Chemograms data) 2 PDX MYC high from the learning cohort (CRCM16 and CRCM04) and 2 PDX MYC high from the validation cohort (CRCM114 and CRCM116). Concerning the MYC low PDX, CRCM05, CRCM10 and CRCM109 were selected from the learning cohort and CRCM112 from the validation cohort respectively.

- Please define "CSCs"

CSC was replaced by cancer stem cells

- Please correct "transcriptionnal"

In page 6, the correction has been applied.

- Please adapt the y axis of S2

The Y axis in the S2 figure was adapted.

3rd Editorial Decision

13 January 2017

Editor requested editorial changes.

3rd Revision - authors' response

25 January 2017

Authors made requested editorial changes.

Corresponding Author Name: Nelson Dusetti or Juan Iovanna

Manuscript Number: EMM-2016-06975